# TOWARDS GLOBAL INTERACTION EFFICIENCY OF GRAPH NETWORKS

## ABSTRACT

A graph inherently embodies comprehensive interactions among all its nodes when viewed globally. Hence, going beyond existing studies in long-range interactions, which focus on interactions between individual node pairs, we study the interactions in a graph through a global perspective. Traditional GNNs acquire such interactions by leveraging local connectivities through aggregations. While this approach has been prevalent, it has shown limitations, such as under-reaching, and over-squashing. In response, we introduce a global interaction perspective and propose interaction efficiency as a metric for assessing GNN performance. This metric provides a unified insight for understanding several key aspects of GNNs, including positional encodings in Graph Transformers, spectral graph filter expressiveness, over-squashing, and the role of nonlinearity in GNNs. Inspired by the global interaction perspective, we present Universal Interaction Graph Convolution, which exhibits superior interaction efficiency. This new architecture achieves highly competitive performance on a variety of graph-level learning tasks. Code is available at `https://github.com/iclrsubmission-towards/UIGC`.

## 1 INTRODUCTION

In recent years, Graph Neural Networks (GNNs) have undergone rapid development. From the model design perspective, various architectures have been proposed, including Spectral Graph Convolution (Hammond et al., 2011; Shuman et al., 2013; Defferrard et al., 2016), Message-passing (MPNN) (Gilmer et al., 2017), Invariant Graph Network (IGN) (Maron et al., 2018; 2019b), Graph Transformer (Dwivedi & Bresson, 2020; Dwivedi et al., 2021; Ying et al., 2021; Lim et al., 2022; Ma et al., 2023), Graph MLP-Mixer (He et al., 2022), etc. From an analytical standpoint, researchers have investigated various factors that limit the performance of GNNs. These factors include over-smoothing (Li et al., 2018; Oono & Suzuki, 2020; Cai & Wang, 2020; Huang et al., 2020; Zhao & Akoglu, 2020), over-squashing or the inability to resolve long-range interaction (Alon & Yahav, 2021; Topping et al., 2021; Liu et al., 2022; Black et al., 2023), graph filter expressiveness (He et al., 2021; Yang et al., 2022a; Wang & Zhang, 2022), and topology expressiveness (Xu et al., 2019; Morris et al., 2019; Maron et al., 2019a; Sato, 2020; Zhang et al., 2022), among others.

However, most GNNs struggle to effectively represent diverse global interactions, and this issue is not well explained by existing analytical tools. In a graph, nodes can have mutual influence with each other, and the influence among distinct pairs of nodes is not independent. Based on this, we adopt a global interaction perspective, where we consider all node pairs and their interactions simultaneously, as illustrated in Figure 1. The dimensions of the interaction space are related to the number of pairs. However, existing GNNs suffer from limited interaction expressiveness, restricting them from representing diverse interactions required by the given tasks. For example, a $k$-layer message-passing GNN fails to represent interactions beyond $k$-hop connectivities. Although Graphormer (Ying et al., 2021; Luo et al., 2022) with fully connected architectures has no such restriction, it directly encodes shortest path distance (SPD) as the structure bias. Recent research by Yang et al. (2023) interprets interactions from a spectral perspective. However, this spectral perspective restricts the exploration of more flexible interactions. Additionally, Ma et al. (2023) demonstrates that their proposed positional encoding can approximate several known interaction patterns simultaneously, but there is a lack of a general understanding of interaction expressiveness.

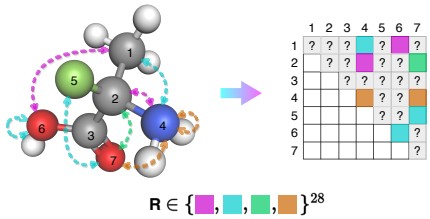

$R \in \{\blacksquare, \blacksquare, \blacksquare, \blacksquare\}^{28}$

Figure 1: Given a graph, each node can interact with all other nodes, including itself. The interaction states between each node pair can be represented by a variable with a continuous or discrete domain. For a graph with $n$ nodes, assuming there are $K$ discrete interaction states, the number of possible interactions is $K^{\frac{n \times (n+1)}{2}}$. In the provided example, the number of interactions is $4^{28}$.

Black et al. (2023) studies global interaction properties with the total effective resistance metric which does not involve the expressiveness analysis.

Interaction expressiveness shares a similar concern with graph filter expressiveness (Balcilar et al., 2021). For a graph with $n$ nodes, the corresponding Laplacian involves $n$ graph Fourier bases, each associated with a filtering coefficient. The concurrent consideration of filtering across different bases results in a filter space of $\mathbb{R}^n$ (He et al., 2021; Yang et al., 2022a; Wang & Zhang, 2022; Bo et al., 2022). Recent studies emphasize the necessity to enhance filter expressiveness to effectively span and cover this space. In a parallel analogy, within the framework of global interactions, each node pair is assigned an interaction coefficient. To view all pair interactions simultaneously, the dimensions of the interaction space are related to the number of pairs.

In essence, the challenge lies in how a model can universally approximate any interactions. To this end, we propose leveraging the Jacobian matrix of all node features before and after the GNN operation as a measurement to quantify interaction efficiency. This approach allows us to differentiate between interaction sensitivity—measuring the sensitivity of model outputs to perturbations in input node features—and interaction expressiveness—quantifying the range of interactions a model can express. The analysis of interaction efficiency expands upon existing studies of over-squashing or long-range interaction by considering dependencies among all pairs of nodes within a graph simultaneously, thus revealing insights into the limitations of GNNs that individual pair-wise analyses cannot encompass. Our contributions are as follows:

- We propose modeling and measuring global interactions with the Jacobian among all node features, utilizing it to assess GNN performance and systematically illustrate interaction limitations in existing GNN designs;
- We demonstrate how the analysis of interaction efficiency offers a comprehensive perspective that generalizes several key findings from GNN studies;
- We present a novel GNN with superior interaction efficiency.

## 2 PRELIMINARIES

Let $G = (\mathcal{V}, \mathcal{E})$ be an undirected graph with vertex set $\mathcal{V}$ of size $n$ and edge set $\mathcal{E}$. $A \in \mathbb{R}^{n \times n}$ is an adjacency matrix with degree matrix $D = \text{diag}(A\mathbf{1}_n)$. Let $\tilde{A} = A + I$ and $\tilde{D} = D + I$, then $\hat{A} = \tilde{D}^{-\frac{1}{2}}\tilde{A}\tilde{D}^{-\frac{1}{2}}$ and $\hat{L} = I - \hat{A}$ refer to the normalized adjacency and Laplacian matrices, respectively. Let $Z \in \mathbb{R}^{n \times d}$ be a $d$-channel graph signal assigned on $G$, also known as a $d$-dimensional node feature matrix, where $Z_{:,i} \in \mathbb{R}^n$ refers to the $i$-th channel of the signal, and $Z_{i,:} \in \mathbb{R}^d$ refers to the signal or feature of the node $i$. $[n]$ denotes the set $\{0, 1, 2, \ldots, n\}$. Given a matrix $M \in \mathbb{R}^{m \times n}$, we denote $\text{vec}(M) \in \mathbb{R}^{mn}$ the vectorization of $M$ and $[M_i]_{i \in [o]} \in \mathbb{R}^{m \times n \times o}$ the concatenation of $[M_1, M_2, \ldots, M_o]$.

### 2.1 GRAPH AUTOMORPHISM

An *automorphism* of a graph is a form of symmetry in which the graph is mapped onto itself while preserving edge–vertex connectivity. Formally, for a graph $G = (\mathcal{V}, \mathcal{E})$, an automorphism is a

permutation $\pi$ of the vertex set $\mathcal{V}$ such that for all vertices $i, j \in \mathcal{V}$, $(i, j) \in \mathcal{E}$ if and only if $(\pi(i), \pi(j)) \in \mathcal{E}$. In other words, $\pi(\mathcal{V})$ is a graph isomorphism from $G$ to itself.

## 2.2 GRAPH CONVOLUTION

Despite the existence of various graph convolution networks, such as ChebyNet (Defferrard et al., 2016), GCN (Kipf & Welling, 2017), CayleyNet (Levie et al., 2019), SGC (Wu et al., 2019), Bern-Net (He et al., 2021), GPR (Chien et al., 2021), JacobiConv (Wang & Zhang, 2022), Corr-free (Yang et al., 2022a), etc., the graph convolution computation can always be generalized into the following form:

$$Z' = f_{\mathcal{W}}(g_{\boldsymbol{\theta}}(\hat{L})Z), \tag{1}$$

where $Z$ is the input graph signals (or node features), $g_{\boldsymbol{\theta}}(\hat{L})$ is the polynomial of $\hat{L}$ with coefficient $\boldsymbol{\theta}$, and $f_{\mathcal{W}}$ is a feature transformation neural network with the set of learnable parameters $\mathcal{W}$. [1]

## 3 INTERACTION EFFICIENCY ANALYSIS

The graph convolution computation in Equation 1 is a differentiable function of inputs $Z$ as both $g_{\boldsymbol{\theta}}$ and $f_{\mathcal{W}}$ are differentiable. Hence, we propose utilizing the Jacobian matrix among all nodes as a formal way to characterize the interactions. Specifically,

**Definition 1** (Global Interaction). *For a GNN computation, given the $a$-th channel of inputs $Z$ and the $b$-th channel of outputs $Z'$, the global interaction among all nodes modeled by the GNN is* $\mathbf{J}^{(a,b)} = \frac{\partial Z'_{:,b}}{\partial Z_{:,a}} \in \mathbb{R}^{n \times n}$.

According to the definition, given the graph $\hat{L}$ and the input node features $Z$, the global interaction of the graph convolution computation can be parameterized by the learnable polynomial coefficients $\boldsymbol{\theta}$ and feature transformation parameters $\mathcal{W}$:

$$\mathbf{J}_{\boldsymbol{\theta},\mathcal{W}} := \mathbf{J}_{\boldsymbol{\theta},\mathcal{W}}^{(a,b)} = \frac{\partial Z'_{:,b}}{\partial Z_{:,a}} = \operatorname{diag}_{i \in [n]} \left( \frac{\partial f_{\mathcal{W}}(K_{i,:})_b}{\partial K_{i,a}} \right) g_{\boldsymbol{\theta}}(\hat{L}) \in \mathbb{R}^{n \times n}, \tag{2}$$

where $K = g_{\boldsymbol{\theta}}(\hat{L})Z \in \mathbb{R}^{n \times d}$. The derivation is given in Appendix A.2. In comparison to existing over-squashing as well as long-range interaction studies that also consider the Jacobian of node features but within an individual node pair (Xu et al., 2018; Alon & Yahav, 2021; Topping et al., 2021; Liu et al., 2022; Black et al., 2023), the global interaction $\mathbf{J}_{\boldsymbol{\theta},\mathcal{W}}$ views all node pairs within a graph simultaneously.

Next, we study the interaction efficiency from the perspective of the interaction sensitivity and the interaction expressiveness respectively, the results of which indicate the limitations and potential solutions in existing graph convolution design.

### 3.1 INTERACTION SENSITIVITY

The determinant $|\mathbf{J}_{\boldsymbol{\theta},\mathcal{W}}| = |\frac{\partial Z'_{:,b}}{\partial Z_{:,a}}|$ serves as a measure of sensitivity, quantifying how the convolution outputs respond to changes in the inputs. A smaller $|\mathbf{J}_{\boldsymbol{\theta},\mathcal{W}}|$ shows that the $b$-th output channel $Z'_{:,b}$ is less likely to be affected by the $a$-th input channel $Z_{:,a}$.

**Proposition 1.** *If $f_{\mathcal{W}}$ is $\alpha$-Lipschitz continuous, the determinant of $\mathbf{J}_{\boldsymbol{\theta},\mathcal{W}}$ is upper bounded as follows:*

$$|\mathbf{J}_{\boldsymbol{\theta},\mathcal{W}}| \leq \alpha^n \left| \prod_{i=1}^{n} g_{\boldsymbol{\theta}}(\boldsymbol{\lambda}_i) \right|, \tag{3}$$

*where $\boldsymbol{\lambda} = \{\lambda_1, \lambda_2, \ldots, \lambda_n\}$ is the spectrum of $\hat{L}$.*

---

[1]Some existing work generalize graph convolution layer into the form $Z^{(l+1)} = \phi_l(g_l(\hat{L})\psi_l(Z^{(l)}))$ (Yang et al., 2022a; Wang & Zhang, 2022). It is equivalent to our $Z^{(l+1)} = f_l(g_l(\hat{L})Z^{(l)})$ by letting $f_l = \psi_l \circ \phi_{l-1}$ when considering over different layers. Please refer to Appendix A.1 for more details.

Note that in most GNNs, $f_{\mathcal{W}}$ is implemented as a one-layer perceptron such as GCN (Kipf & Welling, 2017), GCNII (Ming Chen et al., 2020), SGC (Wu et al., 2019), SSGC (Zhu & Koniusz, 2020), etc., where the resulting graph convolution is then rewritten as $Z' = \sigma(g_{\boldsymbol{\theta}}(\hat{L})ZW)$, where $W$ is a learnable feature transformation matrix and $\sigma$ is a nonlinear activation function. Then the corresponding $\mathbf{J}_{\boldsymbol{\theta},W} = W_{a,b}\mathrm{diag}_{i\in[n]}(\frac{d\sigma(\gamma_i)}{d\gamma_i})g_{\boldsymbol{\theta}}(\hat{L})$, where $\boldsymbol{\gamma} = g_{\boldsymbol{\theta}}(\hat{L})ZW_{:,b}$. According to Proposition 1, we have $|\mathbf{J}_{\boldsymbol{\theta},W}| \leq |(\alpha W_{a,b})^n \prod_{i=1}^{n} g_{\boldsymbol{\theta}}(\boldsymbol{\lambda}_i)|$ with $\sigma$ being $\alpha$-Lipschitz continuous. Detailed derivations are given in Appendix A.4. However, most applied activation functions have $\alpha = \sup_{x\in\mathbb{R}}|\frac{d\sigma(x)}{dx}| \leq 1$. For example, $|\frac{d\mathrm{ReLU}(x)}{dx}| = 0$ or $1$, $|\frac{d\mathrm{Sigmoid}(x)}{dx}| = \mathrm{Sigmoid}(x)(1 - \mathrm{Sigmoid}(x)) \leq 0.25$ and $|\frac{d\mathrm{Tanh}(x)}{dx}| = 1 - \mathrm{Tanh}^2(x) \leq 1$. And the degree normalization operation in GNNs, i.e., $D^{-\frac{1}{2}}AD^{-\frac{1}{2}}$, further shrinks the spectrum, making $\boldsymbol{\lambda}_i$ close to 0. All these factors make $|\mathbf{J}_{\boldsymbol{\theta},W}|$ easily affected by the learnable entry $W_{a,b}$. A small $|W_{a,b}|$ results in $|\mathbf{J}_{\boldsymbol{\theta},W}|$ decaying exponentially with respect to the number of nodes. The determinant of the Jacobian matrix provides a way of quantifying the change of outputs with respect to the perturbation of inputs from a global perspective.

## 3.2 INTERACTION EXPRESSIVENSS

As the interaction in Equation 2 is parameterized by both $\boldsymbol{\theta}$ and $\mathcal{W}$, the interaction space of the graph convolution, i.e., the set of all possible interactions that it can represent, is

$$\mathcal{J}_{\boldsymbol{\theta},\mathcal{W}} = \left\{\mathbf{J}_{\boldsymbol{\theta},\mathcal{W}}\big|\boldsymbol{\theta} \in \mathbb{R}^k, \mathcal{W} \in \mathbb{R}^\star\right\}, \tag{4}$$

where $k$ is the degree of the polynomial, and the dimension of $\mathcal{W} \in \mathbb{R}^\star$ depends on the applied neural network models. To allow comparing the interactions of two models, we then define interaction expressiveness.

**Definition 2** (Interaction Expressiveness). *For two graph models* $\mathsf{GM}_1$ *and* $\mathsf{GM}_2$ *with* $\mathcal{J}_1, \mathcal{J}_2$, *respectively,* $\mathsf{GM}_1$ *is more expressive in interactions than* $\mathsf{GM}_2$, *denoted by* $\mathsf{GM}_1 \succeq \mathsf{GM}_2$, *if and only if* $\mathcal{J}_1 \supseteq \mathcal{J}_2$. *Meanwhile, if* $\mathcal{J}_1 = \mathcal{J}_2$, *we say* $\mathsf{GM}_1$ *is equivalent to* $\mathsf{GM}_2$, *denoted by* $\mathsf{GM}_1 = \mathsf{GM}_2$.

Note that in real-world graph data, symmetric parts of a graph often exhibit identical properties. In alignment with this, general GNNs consistently learn the same interactions for symmetry parts, which we call *symmetry bias*. To show this, we first define node/pair-symmetry as follows.

**Definition 3** (Node/Pair-symmetry). *For a graph $G$ and node features $Z$, two nodes $i, j$ are symmetric, denoted by $i \sim j$, if there exists an automorphism $\pi$ such that (i) $Z_{\pi(i),:} = Z_{i,:}$ for any $i \in [n]$, and (ii) $\pi(i) = j$. Meanwhile, two pairs $(i, j)$, $(k, l)$ are symmetric, denoted by $(i, j) \sim (k, l)$, if $i \sim k$ and $j \sim l$ are under the same automorphism.*

According to the definition, pair-symmetry is an equivalence relation, thus we can get the quotient space $\{(i, j)|i, j \in [n]\}/\sim$, i.e., the partitions of pairs based on their pair-symmetry. Examples are given in Figure 2. Graphs (a) to (d) have the same number of nodes, but different topologies result in different pair partitions based on pair-symmetry, where more symmetric graphs generally have less number of partitions. Also, in graphs with diverse node features, the partitions are further decided by node differences as the comparisons between graph (d) and (e), which means non-symmetric pairs would not be distinguished without considering node differences. Node-symmetry under graph automorphism is also studied by Wang & Zhang (2022); Xu et al. (2021). We further extend it to pair-symmetry, which serves as the basic concept in the interaction expressiveness analysis.

**Proposition 2.** *For any $\mathbf{J} \in \mathcal{J}$ in the graph convolution and any $(i, j)$, $(k, l)$ within the same pair partition, i.e. $(i, j) \sim (k, l)$, we have $\mathbf{J}_{i,j} = \mathbf{J}_{k,l}$, $Z'_i = Z'_k$ and $Z'_j = Z'_l$.*

Proposition 2 shows that the graph convolution always learns the same interactions for symmetric pairs, i.e. symmetry bias. Hence we only need to consider the interactions over different pair partitions instead of individual pairs. This dramatically reduces the interaction space in Eq.4. Let $\eta = |\{(i, j)|i, j \in [n]\}/\sim|$ be the number of pair partitions, and $\mathcal{J}^\sim \subseteq \mathbb{R}^\eta$ be the interaction space over different partitions. The model with $\mathcal{J}^\sim = \mathbb{R}^\eta$ corresponds to the highest expressiveness, i.e., it is powerful enough to model any interaction with proper neural parameter assignments. We call it *universal expressiveness* of interactions. Unfortunately, existing graph convolution is far from universality as follows.[2]

---

[2]We provide similar interaction expressiveness analysis of MPNN framework in Appendix B due to the page limits.

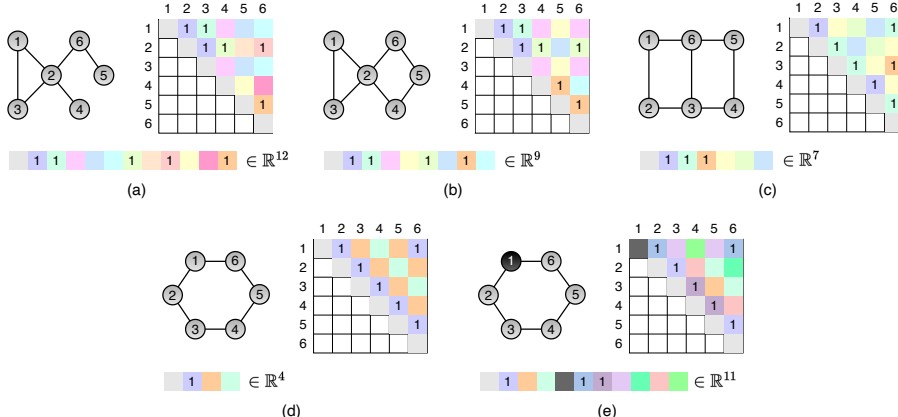

Figure 2: We explore graphs and their pair partitions through the lens of pair-symmetry. Graph (a) to (e) present graphs with 6 nodes. For each of them, there exists a total of 21 pairs. These pairs are visually represented in a matrix on the right-hand side. Entries corresponding to symmetric pairs, as defined in Definition 3, are highlighted in the same color, and entries corresponding to direct edges are marked with '1'. Below each graph, a vector summarizes the number of pair partitions in that particular graph.

**Proposition 3.** *For the interaction space $\mathcal{J}_{\boldsymbol{\theta},\mathcal{W}}$ of the graph convolution, we have* $\text{vec}(\mathcal{J}_{\boldsymbol{\theta},\mathcal{W}}) \subseteq \{I \otimes \text{diag}(\boldsymbol{\beta})[\text{vec}(U_i \otimes U_i)]_{i \in [n]}\boldsymbol{\alpha} \big| \boldsymbol{\alpha}, \boldsymbol{\beta} \in \mathbb{R}^n\}$, *where* $\hat{L} = U\Lambda U^\top$.

$\otimes$ is the Kronecker product. Proposition 3 shows that $\mathcal{J}_{\boldsymbol{\theta},\mathcal{W}}$ is an at most $2n$ dimensional subspace within $\mathbb{R}^\eta$, which is far from universal expressiveness of interactions.

Next, the objective is to improve the interaction expressiveness of graph convolutions. To make it more general, in the following analysis, we extend $g_{\boldsymbol{\theta}}(\hat{L})$ in graph convolution to any set of graph matrix representations $\mathcal{S} \subseteq \mathbb{R}^{n \times n}$. Correspondingly,

**Proposition 4.** *With* $\mathcal{J}_{S,\mathcal{W}} = \{\text{diag}_{i \in [n]}(\frac{\partial f_{\mathcal{W}}(K_{i,:})_b}{\partial K_{i,a}})S \big| S \in \mathcal{S}, \mathcal{W} \in \mathbb{R}^\star\}$, *we have*

*(i)* $\text{vec}(\mathcal{J}_{S,\mathcal{W}}) \subseteq \bigcup_{S \in \mathcal{S}}\{\text{vec}(\text{diag}(\boldsymbol{\alpha})S)\big|\boldsymbol{\alpha} \in \mathbb{R}^n\}$;
*(ii)* $\text{vec}(\mathcal{S}) \subseteq \bigcup_{S \in \mathcal{S}}\{\text{vec}(\text{diag}(\boldsymbol{\alpha})S)\big|\boldsymbol{\alpha} \in \mathbb{R}^n\}$.

Proposition 4(i) shows that $\mathcal{J}_{S,\mathcal{W}}$ is bounded by $\text{vec}(\mathcal{J}_\mathcal{S}^{\text{upper}}) = \bigcup_{S \in \mathcal{S}}\{\text{vec}(\text{diag}(\boldsymbol{\alpha})S)\big|\boldsymbol{\alpha} \in \mathbb{R}^n\}$. Inspired by this, we simplify the objective of improving the interaction expressiveness of graph convolution by improving its upper bound $\mathcal{J}_\mathcal{S}^{\text{upper}}$, which is parameterized by $\mathcal{S}$. Furthermore, Proposition 4(ii) shows that we can push $\mathcal{J}_\mathcal{S}^{\text{upper}}$ to universality by letting $\mathcal{S}^\sim = \mathbb{R}^\eta$.

## 4 UNIVERSAL INTERACTION GRAPH CONVOLUTION (UIGC)

For input node features with $d'$ channels, the model learns an individual interaction for each channel. We use $S \in \mathbb{R}^{n \times n \times d'}$ to denote all $d'$ interactions over these channels. Then, $S_{i,j,:}$ corresponds to the interaction of all $d'$ channels between the node pair $(i, j)$. Note that any $S$ can be viewed as a mapping $f : \{(i, j)\big|i, j \in [n]\} \mapsto \{S_{i,j,:}\big|i, j \in [n]\}$, hence we can model any $S$ by approximating the corresponding mapping $f$.

**Proposition 5.** *Given a MLP* $f_\Theta : \mathbb{R}^d \mapsto \mathbb{R}^{d'}$ *with learnable parameters* $\Theta \in \mathbb{R}^\star$, *for any* $S \in \mathbb{R}^{n \times n \times d'}$, *injective mapping* $\varphi : \mathbb{R}^2 \mapsto \mathbb{R}^d$ *and* $\epsilon > 0$, *there exists* $\Theta_0$ *such that*

$$\sup_{i,j \in [n]} \big\| f_\Theta \circ \varphi((i, j))|_{\Theta = \Theta_0} - S_{i,j,:}\big\| < \epsilon. \tag{5}$$

Proposition 5 shows the general result regarding the potential interaction expressiveness a model can achieve without specifying symmetry properties. Then, to ensure symmetry bias, we relax the

injectivity of $\varphi$ by ensuring that symmetric pairs always share the same output as follows.

$$\varphi^{\text{LE}} : \{\text{Local structures and/or node/edge features of } (i,j) | i,j \in [n]\} \mapsto \mathbb{R}^d. \tag{6}$$

$\varphi^{\text{LE}}$ acts as a local encoding operation. Then, the complete of UIGC layer is

$$E = \left[ \varphi^{\text{LE}}((i,j)) \right]_{i,j \in [n]} \in \mathbb{R}^{n \times n \times d}$$

$$\mathbf{J} = [f_\Theta(E_{i,j,:})]_{i,j \in [n]} \in \mathbb{R}^{n \times n \times d'} \tag{7}$$

$$Z' = f_{\mathcal{W}} \left( [\mathbf{J}_{:,:,i} Z_{:,i}]_{i \in [d]} \right) \in \mathbb{R}^{n \times d}.$$

As $\varphi^{\text{LE}}$ restricts the pair encodings to the mapping of local structures, it produces identical encodings for pairs within the same partitions. The universal expressiveness of interactions across different partitions is preserved if $\varphi^{\text{LE}}$ is injective to these partitions. However, achieving injectivity is generally challenging. Different implementations of $\varphi^{\text{LE}}$ exhibit varying discrimination abilities on partitions. For a set of implementations of $\varphi^{\text{LE}}$, their discrimination abilities form a partial order. In this partial order, the one injective to the partitions is the most discriminative. Consequently, we consolidate diverse $\varphi^{\text{LE}}$ implementations in existing GNNs into a Directed Acyclic Graph (DAG) and compare their discrimination abilities. For detailed information, please refer to Appendix C. Finally, ignoring the approximation error $\epsilon$ of MLP, the universal expressiveness of interactions holds as $\{[f_\Theta \circ \varphi^{\text{LE}}((i,j))]_{i,j \in [n]} | \Theta \in \mathbb{R}^\star\} = \mathcal{S}^\sim = \mathbb{R}^\eta$.

***Revisiting positional encodings in Graph Transformers via the lens of global interactions.*** Some positional encoding studies in Graph Transformers can be viewed as implementations of $\varphi^{\text{LE}}$ with various discrimination abilities of non-symmetric pairs. For example, the shortest path distance matrix (SPD) used in Ying et al. (2021); Luo et al. (2022) can be viewed as an implementation of $\varphi^{\text{LE}}$, but this encoding strategy cannot distinguish non-symmetric pairs with the same topological distance. Ma et al. (2023) proposes relative random walk probabilities (RRWP) positional encoding in Graph Transformers and shows its great expressiveness that can approximate SPD and others by combining with MLP. In general, since there always exists a mapping from a local encoding to a less discriminative one, we can approximate any desired interaction, e.g. SPD, with any more discriminative local encoding as the inputs of MLP. Similarly, a weak local encoding will result in the bottleneck of approximating complex interactions as $f_\Theta \circ \varphi^{\text{LE}}((i,j)) \preceq \varphi^{\text{LE}}((i,j))$ according to the partial order of the discrimination ability. Consistently, the adjacency matrix only encodes each pair according to the explicit 1-hop connections, which means it cannot distinguish all pairs with (or without) 1-hop connections. In comparison, normalized adjacency $D^{-\frac{1}{2}} A D^{-\frac{1}{2}}$ is a stronger one, as it further distinguishes connected pairs with different degrees. Other interesting local encoding strategies include GSO (Sandryhaila & Moura, 2013; Dasoulas et al., 2021), Spectrum-smoothness (Yang et al., 2022a), etc. Although these local encoding strategies are motivated by different theories, they play a similar role in identifying non-symmetric pairs, such as positional encodings in Graph Transformers and polynomial bases in spectral graph filters. Generally, a more discriminative one is more helpful. For example, higher degree polynomials with better filter expressiveness, as recommended in some works (Wang & Zhang, 2022; Yang et al., 2022a), are also more discriminative. More comparisons are given in Appendix C.

***Revisiting the role of nonlinearity as well as spectral graph filter expressiveness via the lens of global interactions.*** Nonlinearity plays an essential role in interaction expressiveness, as shown in our UIGC, where the effects of nonlinearity are analogous to that in achieving universal approximation power of MLP (Cybenko, 1989; Hornik et al., 1989). While without nonlinearity, the interaction computation degrades into the linear combinations of a group of bases as that in spectral graph filters approximated with polynomials. Generally, improving polynomials degree can improve filter/interaction expressiveness as $\{g_{\boldsymbol{\theta}}(\Lambda) | \boldsymbol{\theta} \in \mathbb{R}^k\} \subseteq \{g_{\boldsymbol{\theta}}(\Lambda) | \boldsymbol{\theta} \in \mathbb{R}^{k+1}\}$ and $\{g_{\boldsymbol{\theta}}(\hat{L}) | \boldsymbol{\theta} \in \mathbb{R}^k\} \subseteq \{g_{\boldsymbol{\theta}}(\hat{L}) | \boldsymbol{\theta} \in \mathbb{R}^{k+1}\}$, where $\hat{L} = U \Lambda U^\top$. Also, it has been proved that improving the polynomial degree to $k = n$ achieves universal expressiveness of filters, i.e. $\{g_{\boldsymbol{\theta}}(\Lambda) | \boldsymbol{\theta} \in \mathbb{R}^n\} = \mathbb{R}^n$ (Yang et al., 2022a; Wang & Zhang, 2022). However, it cannot improve expressiveness by further setting $k > n$ as it is bounded by $\text{vec}(\{g_{\boldsymbol{\theta}}(\hat{L}) | \boldsymbol{\theta} \in \mathbb{R}^k\}) \subseteq \text{span} \left( \{\text{vec}(U_i \otimes U_i) | i \in [n]\} \right)$ for any $k$. This means that the universal expressiveness of filters only corresponds to an $n$-dimensional subspace of the interaction space, which is far from the universal expressiveness of interactions. Thankfully, involving nonlinearity helps to break this bound. Also, a larger $k$ in polynomial filters can easily result in numerical instability, making it less applicable.

***Revisiting over-squashing via the lens of global interactions.*** The over-squashing issue can be understood in terms of one node representation failing to be affected by some other input node features at long distances, which is also known as long-range interaction issue (Alon & Yahav, 2021; Topping et al., 2021; Liu et al., 2022; Black et al., 2023). Topping et al. (2021) and Black et al. (2023) apply the Jacobian of the two node features/representations as a formal way to access the over-squashing effects. Liu et al. (2022) studies the norm of node feature perturbation between two nodes. They all theoretically show that the dependency effects decay exponentially with respect to the distance. However, these studies focus on the case of two given nodes with a known distance in the graph. It corresponds to a local view where the dependency effects are always considered within each pair of nodes individually. Our interaction efficiency analysis provides a global view that studies global interaction sensitivity and interaction expressiveness among all nodes simultaneously, which cannot be identified by the local view.

## 5 EXPERIMENTS

In this section, we first verify the effectiveness of our UIGC in approximating various synthetic interactions. Then, we evaluate UIGC on graph classification and regression datasets.

### 5.1 LEARNING INTERACTIONS ON SYNTHETIC DATA

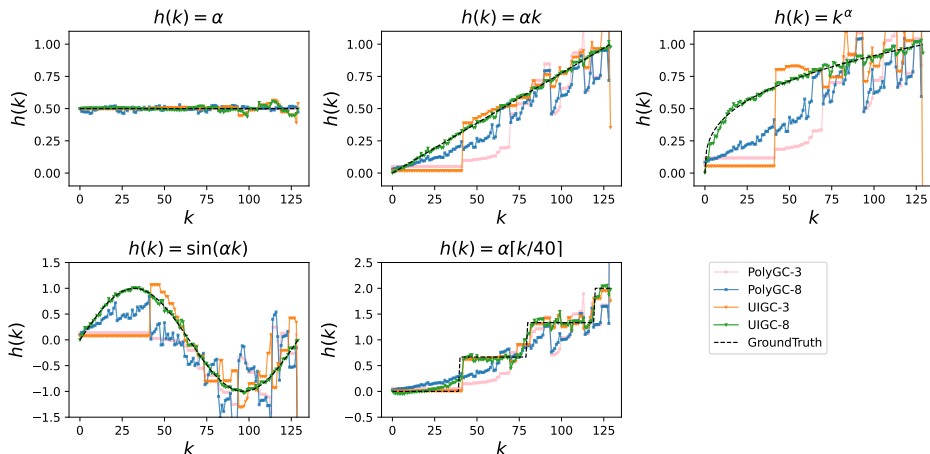

Figure 3: Illustrations of 5 synthetic interaction patterns learned by PolyGC and UIGC. $k$ is the pair partition index. $h(k)$ is the synthetic interaction function. $\alpha$ is a scaling factor controlling output.

**Setup.** We randomly select a graph from TUDataset/NCI1. Based on pair-symmetry, all node pairs form 130 partitions. We index these partitions from 0 to 129 and then assign five distinct interaction patterns to the partition indices with different approximation difficulties, as depicted in Figure 3. We test graph convolution models with polynomial filters, i.e. PolyGC, and our UIGC models. To ensure a fair comparison, UIGC and PolyGC apply the same $\varphi^{\text{LE}}$, i.e. $[\hat{A}^k]_{k \in [K]}$. A larger value of $K$ achieves a more discriminative $\varphi^{\text{LE}}$. For the given graph, $K \geq 8$ is sufficient to distinguish all pair partitions. In our tests, we evaluate cases with $K = 3$ and $K = 8$, representing insufficient and sufficient discrimination abilities of $\varphi^{\text{LE}}$, respectively. In PolyGC, they correspond to 3-degree and 8-degree polynomials. Other values of $K$ can be chosen to demonstrate similar results. Then the corresponding interaction computations of PolyGC and UIGC are $\mathbf{J}_{\boldsymbol{\theta}} = \sum_{k=0}^{K} \boldsymbol{\theta}_k \hat{A}^k$ and $\mathbf{J}_{\boldsymbol{\theta}} = \text{MLP}_{\boldsymbol{\theta}}([\hat{A}^k]_{k \in [K]})$ respectively. The node features $X$ are randomly generated. Finally, the model learns node representations for each node by optimizing $\arg\min_{\boldsymbol{\theta}} \|\mathbf{J}_{\boldsymbol{\theta}} X - \mathbf{J}_{\text{GT}} X\|_2$.

**Result.** When applying fewer bases, both PolyGC-3 and UIGC-3 struggle to approximate the given interactions effectively, resulting in significant errors. Notably, the partitions indexed below 50 cannot be distinguished from each other across all four interaction patterns. This arises from the limited discrimination ability of $[\hat{A}^k]_{k \in [3]}$, which cannot differentiate pairs beyond 3 hops. In contrast, when applying more bases as $[\hat{A}^k]_{k \in [8]}$, both PolyGC-8 and UIGC-8 can distinguish each partition suc-

Table 1: Mean square error of approximations on synthetic interactions.

| Function | $h(k) = \alpha$ | $h(k) = \alpha k$ | $h(k) = k^\alpha$ | $h(k) = \sin(\alpha k)$ | $h(k) = \alpha\lceil k/40 \rceil$ |
|---|---|---|---|---|---|
| PolyGC-3 | 0 | 0.1326 | 0.2567 | 0.5346 | 0.3668 |
| PolyGC-8 | 0.0002 | 0.1709 | 0.3073 | 0.3446 | 0.4523 |
| UIGC-3 | 0.0005 | 0.0143 | 0.1160 | 0.2598 | 0.0184 |
| UIGC-8 | 0.0003 | 0.0002 | 0.0009 | 0.0018 | 0.0111 |

cessfully. However, PolyGC-8 does not exhibit significant improvement with an increased number of bases. This observation suggests that polynomial filters, while performing well in approximating graph filters as demonstrated in previous studies (He et al., 2021; Bo et al., 2022), struggle to effectively approximate interactions. In contrast, UIGC-8 outperforms the others by achieving the best fit on all interaction patterns, with significantly smaller errors as indicated in Table 1. This verifies our analysis that discriminative local encoding combined with MLP achieves the universal approximation ability of interactions. Furthermore, it shows that the piecewise interaction is the most challenging to approximate, while the constant with all partitions sharing the same interaction value is the easiest where all four implementations can fit accurately.

## 5.2 BENCHMARKING UIGC

We evaluate UIGC on datasets from Benchmarking GNN (Dwivedi et al., 2020), OGB (Hu et al., 2020) and TUDataset (Morris et al., 2020) respectively. All baseline results are quoted from their official leaderboards [3] or the original papers.

**Baselines.** The baseline models used for comparisons include: GK (Shervashidze et al., 2009), RW (Vishwanathan et al., 2010), PK (Neumann et al., 2016), FGSD (Verma & Zhang, 2017), AWE (Ivanov & Burnaev, 2018), DGCNN (Zhang et al., 2018), PSCN (Niepert et al., 2016), DCNN (Atwood & Towsley, 2016), ECC (Simonovsky & Komodakis, 2017), DGK (Yanardag & Vishwanathan, 2015), CapsGNN (Xinyi & Chen, 2019), GIN (Xu et al., 2019), $k$-GNN (Morris et al., 2019), IGN (Maron et al., 2018), PPGNN (Maron et al., 2019a), Soft-mask (Yang et al., 2021), GCN$^2$ (de Haan et al., 2020), GraphSage (Hamilton et al., 2017), GAT (Veličković et al., 2018), GatedGCN-PE (Bresson & Laurent, 2017), MPNN (sum) (Gilmer et al., 2017), DeeperGCN (Li et al., 2020), PNA (Corso et al., 2020), DGN (Beani et al., 2021), GSN (Bouritsas et al., 2020), GINE-APPNP (Brossard et al., 2020), PHC-GNN (Le et al., 2021), ExpC (Yang et al., 2022b), GT (Dwivedi et al., 2020), SAN (Kreuzer et al., 2021), Graphormer (Ying et al., 2021), K-Subgraph SAT (Chen et al., 2022), EGT (Hussain et al., 2022), GPS (Rampášek et al., 2022).

Following baseline settings, we use the parameter budgets ∼500k for ZINC, ∼100k for MNIST, and no parameter limitation for ogbg-molpcba. Both ZINC and ogbg-molpcba are small molecule graphs that have sparse connections. General message-passing models rely on connectivity to compute interactions. A sparse-connected graph requires a deeper model to capture the long-range interactions, but this will lead to over-squashing and over-smoothing issues. UIGC infers the interaction of each pair directly through their local encodings, which will not be affected by the connectivity of graphs. The results in Table 2 show that UIGC outperforms baselines on these sparsely connected graphs.

TUDataset involves small-scale datasets. We use the standard 10-fold cross-validation and dataset splits in Zhang et al. (2018), and then report our results following the rule as described in Xu et al. (2019) and Ying et al. (2018). The results are presented in Tab.3. Our tested datasets have a number of graphs ranging from 1000 to 4000. This kind of small-scale graph data can easily result in overfitting, making it less effective in leveraging more learnable parameters, as shown in (Yang et al., 2021). Also, it is unclear whether the popular Graph Transformer-based models can fully show their power on these small-scale data as there are not many results provided. In UIGC, we utilize the MLP $f_\Theta$ to model interactions. The classification improvement shows the alleviation of the overfitting issue. However, datasets like IMDB-B have no classification gains, indicating that label-related interaction patterns on these graphs may be inherently simple and can be easily captured by basic models. Detailed hyperparameter settings and the number of parameters of the model are given in Appendix D.2.

---

[3]https://paperswithcode.com/sota/graph-regression-on-zinc-500k and https://ogb.stanford.edu/docs/leader_graphprop/

Table 2: Results on ZINC, ogbg-molpcba, and MNIST. The best results are in bold, and the second-best are underlined.

| Method | ZINC MAE $\downarrow$ | MNIST ACC(%) $\uparrow$ | ogbg-molpcba AP(%) $\uparrow$ |
|---|---|---|---|
| GCN | 0.367±0.011 | 90.705±0.218 | 24.24±0.34 |
| GIN | 0.526±0.051 | 96.485±0.252 | 27.03±0.23 |
| GAT | 0.384±0.007 | 95.535±0.205 | - |
| GraphSage | 0.398±0.002 | - | - |
| GatedGCN-PE | 0.214±0.006 | 97.340±0.143 | - |
| MPNN | 0.145±0.007 | - | - |
| DeeperGCN | - | - | 28.42±0.43 |
| PNA | 0.142±0.010 | 97.94±0.12 | 28.38±0.35 |
| DGN | 0.168±0.003 | - | 28.85±0.30 |
| GSN | 0.101±0.010 | - | - |
| GINE-APPNP | - | - | 29.79±0.30 |
| PHC-GNN | - | - | 29.47±0.26 |
| ExpC | - | - | 23.42±0.29 |
| GT | 0.226±0.014 | - | - |
| SAN | 0.139±0.006 | - | 27.65±0.42 |
| Graphormer | 0.122±0.006 | - | - |
| K-Subgraph SAT | 0.094±0.008 | - | - |
| EGT | 0.108±0.009 | 98.173±0.087 | - |
| GPS | 0.070±0.004 | 98.051±0.126 | 29.07±0.28 |
| UIGC (Ours) | **0.060±0.002** | **98.272±0.119** | **30.24±0.27** |

Table 3: Results on TUDataset.

| Method | ENZYMES | NCI1 | NCI109 | PTC_MR | PROTEINS | IMDB-B | RDT-B |
|---|---|---|---|---|---|---|---|
| GK | 32.70±1.20 | 62.49±0.27 | 62.35±0.3 | 55.65±0.5 | 71.39±0.3 | - | 77.34±0.18 |
| RW | 24.16±1.64 | - | - | 55.91±0.3 | 59.57±0.1 | - | - |
| PK | - | 82.54±0.5 | - | 59.5±2.4 | 73.68±0.7 | - | - |
| FGSD | - | 79.80 | 78.84 | 62.8 | 73.42 | 73.62 | - |
| AWE | 35.77±5.93 | - | - | - | - | 74.45±5.80 | 87.89±2.53 |
| DGCNN | 51.0±7.29 | 74.44±0.47 | - | 58.59±2.5 | 75.54±0.9 | 70.03±0.90 | - |
| PSCN | - | 74.44±0.5 | - | 62.29±5.7 | 75±2.5 | 71±2.3 | 86.30±1.58 |
| DCNN | - | 56.61±1.04 | - | - | 61.29±1.6 | 49.06±1.4 | - |
| ECC | 45.67 | 76.82 | 75.03 | - | - | - | - |
| DGK | 53.43±0.91 | 80.31±0.46 | 80.32±0.3 | 60.08±2.6 | 75.68±0.5 | 66.96±0.6 | 78.04±0.39 |
| GraphSAGE | 58.2±6.0 | 76.0±1.8 | - | - | - | 72.3±5.3 | - |
| CapsGNN | 54.67±5.67 | 78.35±1.55 | - | - | 76.2±3.6 | 73.1±4.8 | - |
| GIN | - | 82.7±1.7 | - | 64.6±7.0 | 76.2±2.8 | **75.1±5.1** | 92.4±2.5 |
| $k$-GNN | - | 76.2 | - | 60.9 | - | 74.2 | - |
| IGN | - | 74.33±2.71 | 72.82±1.45 | 58.53±6.86 | 76.58±5.49 | 72.0±5.54 | - |
| PPGNN | - | 83.19±1.11 | 82.23±1.42 | 66.17±6.54 | 77.20±4.73 | 73.0±5.77 | - |
| Soft-mask | 60.3±5.26 | 83.3±1.88 | - | - | 76.8±4.15 | 75.0±5.95 | 93.1±2.25 |
| GCN$^2$ | - | 82.74±1.35 | 83.00±1.89 | 66.84±1.79 | 71.71±1.04 | 74.80±2.01 | - |
| UIGC (Ours) | **74.83±7.17** | **85.09±1.12** | **83.35±0.87** | **67.47±7.76** | **77.27±4.33** | 74.90±3.24 | **93.40±1.09** |

## 6 CONCLUSION

We propose to study the global interactions of a graph as a way to access GNN performance. To this end, we use the Jacobian of all node features before and after GNN operations as the interaction efficiency metric. It can be applied to various graph models and also provides new interpretations into several widely studied issues.

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

## A  PROOFS AND DERIVATIONS

### A.1  UNIFIED GRAPH CONVOLUTION

$$
\begin{aligned}
H^{(l+1)} &= \phi_l(g_l(\hat{L})\psi_l(H^{(l)})) \\
&= \phi_l(g_l(\hat{L})\psi_l \circ \phi_{l-1}(g_{l-1}(\hat{L})(\dots \psi_1 \circ \phi_0(g_0(\hat{L})\psi_0(X)))))
\end{aligned}
\tag{8}
$$

Let $f_l = \psi_l \circ \phi_{l-1}$, then the $K$-layer graph convolution operation can be represented as

$$
\begin{aligned}
Z^{(1)} &= \psi_0(X) \\
Z^{(l+1)} &= f_l(g_l(\hat{L})Z^{(l)}) \\
Y &= \phi_{K-1}(g_{K-1}(\hat{L})Z^{(K-1)}) = \psi_K^{-1}(Z^{(K)}),
\end{aligned}
\tag{9}
$$

where we suppose $\psi_K$ is invertible.

### A.2  DERIVATIONS OF EQUATION 2

With

$$
Z' = f_{\mathcal{W}}(g_{\boldsymbol{\theta}}(\hat{L})Z) = \begin{pmatrix} f_{\mathcal{W}}\left(g_{\boldsymbol{\theta}}(\hat{L})_{1,:}Z\right) \\ \vdots \\ f_{\mathcal{W}}\left(g_{\boldsymbol{\theta}}(\hat{L})_{n,:}Z\right) \end{pmatrix} \in \mathbb{R}^{n \times d},
\tag{10}
$$

we have

$$
\begin{aligned}
Z'_{i,b} &= f_{\mathcal{W}}\left(g_{\boldsymbol{\theta}}(\hat{L})_{i,:}Z\right)_b \\
&= f_{\mathcal{W}}\left(\sum_{j=1}^{n} g_{\boldsymbol{\theta}}(\hat{L})_{i,j}Z_{j,:}\right)_b \\
&= f_{\mathcal{W}}\left(\sum_{j=1}^{n} g_{\boldsymbol{\theta}}(\hat{L})_{i,j}Z_{j,1}, \dots, \sum_{j=1}^{n} g_{\boldsymbol{\theta}}(\hat{L})_{i,j}Z_{j,a}, \dots\right)_b.
\end{aligned}
\tag{11}
$$

Then we have

$$
\frac{\partial Z'_{i,b}}{\partial Z_{j,a}} = \frac{\partial f_{\mathcal{W}}(K_{i,:})_b}{\partial K_{i,a}} g_{\boldsymbol{\theta}}(\hat{L})_{i,j} \in \mathbb{R}
\tag{12}
$$

among which $K = g_{\boldsymbol{\theta}}(\hat{L})Z \in \mathbb{R}^{n \times d}$ and $K_{u,v} = \sum_{i=1}^{n} g_{\boldsymbol{\theta}}(\hat{L})_{u,i}Z_{i,v} \in \mathbb{R}$. Then

$$
\frac{\partial Z'_{i,b}}{\partial Z_{:,a}} = \frac{\partial f_{\mathcal{W}}(K_{i,:})_b}{\partial K_{i,a}} g_{\boldsymbol{\theta}}(\hat{L})_{i,:} \in \mathbb{R}^n.
\tag{13}
$$

Finally,

$$
\mathbf{J}_{\boldsymbol{\theta},\mathcal{W}} = \frac{\partial Z'_{:,b}}{\partial Z_{:,a}} = \operatorname{diag}_{i\in[n]}\left(\frac{\partial f_{\mathcal{W}}(K_{i,:})_b}{\partial K_{i,a}}\right) g_{\boldsymbol{\theta}}(\hat{L}) \in \mathbb{R}^{n \times n}.
\tag{14}
$$

### A.3 PROOF OF PROPOSITION 1

*Proof.*

$$
\begin{aligned}
|\mathbf{J}_{\boldsymbol{\theta},\mathcal{W}}| &= \left| \text{diag}_{i\in[n]} \left( \frac{\partial f_{\mathcal{W}}(K_{i,:})_b}{\partial K_{i,a}} \right) g_{\boldsymbol{\theta}}(\hat{L}) \right| \\
&= \left| \text{diag}_{i\in[n]} \left( \frac{\partial f_{\mathcal{W}}(K_{i,:})_b}{\partial K_{i,a}} \right) \right| \left| g_{\boldsymbol{\theta}}(\hat{L}) \right| \\
&= \prod_{i=1}^{n} \frac{\partial f_{\mathcal{W}}(K_{i,:})_b}{\partial K_{i,a}} \prod_{i=1}^{n} g_{\boldsymbol{\theta}}(\boldsymbol{\lambda}_i) \\
&\leq \alpha^n \left| \prod_{i=1}^{n} g_{\boldsymbol{\theta}}(\boldsymbol{\lambda}_i) \right| \quad / * \text{As} \sup_{\boldsymbol{x}\in\mathbb{R}^d} \left| \frac{\partial f_{\mathcal{W}b}}{\partial \boldsymbol{x}_a} \right| = \alpha * /
\end{aligned}
\tag{15}
$$

$\square$

### A.4 DERIVATIONS OF ONE-LAYER PERCEPTRON IMPLEMENTATION

With $Z' = \sigma(g_{\boldsymbol{\theta}}(\hat{L})ZW)$, we have,

$$
\begin{aligned}
Z'_{i,b} &= \sigma \left( g_{\boldsymbol{\theta}}(\hat{L})_{i,:} Z W_{:,b} \right) \\
&= \sigma \left( g_{\boldsymbol{\theta}}(\hat{L})_{i,:} \sum_{u=1}^{d} Z_{:,u} W_{u,b} \right) \\
&= \sigma \left( \sum_{v=1}^{n} \sum_{u=1}^{d} g_{\boldsymbol{\theta}}(\hat{L})_{i,v} Z_{v,u} W_{u,b} \right).
\end{aligned}
\tag{16}
$$

Then we have

$$
\frac{\partial Z'_{i,b}}{\partial Z_{j,a}} = \frac{d\sigma(\boldsymbol{\gamma}_i)}{d\boldsymbol{\gamma}_i} g_{\boldsymbol{\theta}}(\hat{L})_{i,j} W_{a,b} \in \mathbb{R}
\tag{17}
$$

among which $\boldsymbol{\gamma}_i = g_{\boldsymbol{\theta}}(\hat{L})_{i,:} Z W_{:,b} \in \mathbb{R}$. Then

$$
\frac{\partial Z'_{i,b}}{\partial Z_{:,a}} = \frac{d\sigma(\boldsymbol{\gamma}_i)}{d\boldsymbol{\gamma}_i} g_{\boldsymbol{\theta}}(\hat{L})_{i,:} W_{a,b} \in \mathbb{R}^n.
\tag{18}
$$

Finally,

$$
\mathbf{J}_{\boldsymbol{\theta},W} = \frac{\partial Z'_{:,b}}{\partial Z_{:,a}} = \text{diag}_{i\in[n]} \left( \frac{d\sigma(\boldsymbol{\gamma}_i)}{d\boldsymbol{\gamma}_i} \right) g_{\boldsymbol{\theta}}(\hat{L}) W_{a,b} \in \mathbb{R}^{n\times n}.
\tag{19}
$$

Correspondingly,

$$
\begin{aligned}
|\mathbf{J}_{\boldsymbol{\theta},W}| &= \left| W_{a,b} \text{diag}_{i\in[n]} \left( \frac{d\sigma(\boldsymbol{\gamma}_i)}{d\boldsymbol{\gamma}_i} \right) g_{\boldsymbol{\theta}}(\hat{L}) \right| \\
&= W_{a,b}^n \left| \text{diag}_{i\in[n]} \left( \frac{d\sigma(\boldsymbol{\gamma}_i)}{d\boldsymbol{\gamma}_i} \right) \right| \left| g_{\boldsymbol{\theta}}(\hat{L}) \right| \\
&= W_{a,b}^n \prod_{i=1}^{n} \frac{d\sigma(\boldsymbol{\gamma}_i)}{d\boldsymbol{\gamma}_i} \prod_{i=1}^{n} g_{\boldsymbol{\theta}}(\boldsymbol{\lambda}_i) \\
&\leq \left| (\alpha W_{a,b})^n \prod_{i=1}^{n} g_{\boldsymbol{\theta}}(\boldsymbol{\lambda}_i) \right| \quad / * \text{As} \sup_{x\in\mathbb{R}} \left| \frac{d\sigma(x)}{dx} \right| = \alpha * /
\end{aligned}
\tag{20}
$$

### A.5 PROOF OF PROPOSITION 2

For a graph automorphism $\pi$, we use $P_\pi$ to denote the corresponding permutation matrix of $\pi$.

**Lemma 1.** *Given a graph $G$ and node features $Z$, both $Z'$ and $\mathbf{J}$ in the graph convolution are invariant to graph automorphism, i.e. for any automorphism $\pi$ of $G$ with $P_\pi Z = Z$, we have $P_\pi Z' = Z'$ and $P_\pi \mathbf{J} P_\pi^\top = \mathbf{J}$.*

*Proof.* According to the definition of graph automorphism, we have $P_\pi \hat{L} P_\pi^\top = \hat{L}$ and $P_\pi \hat{L}^2 P_\pi^\top = \hat{L}^2$. Then

$$
\begin{aligned}
P_\pi g_{\boldsymbol{\theta}}(\hat{L}) P_\pi^\top &= P_\pi \left( \sum_{i=0}^k \alpha_i L^i \right) P_\pi^\top \\
&= \sum_{i=0}^k \alpha_i P_\pi L^i P_\pi^\top \\
&= \sum_{i=0}^k \alpha_i L^i \\
&= g_{\boldsymbol{\theta}}(\hat{L}).
\end{aligned}
\tag{21}
$$

Let $S = g_{\boldsymbol{\theta}}(\hat{L})$ for the representation simplicity. We have $P_\pi S P_\pi^\top = S$ and $P_\pi S Z = P_\pi S P_\pi^\top P_\pi Z = SZ$. Then

$$
\begin{aligned}
P_\pi Z' &= P_\pi f_{\mathcal{W}}(SZ) \\
&= f_{\mathcal{W}}(P_\pi SZ) \\
&= f_{\mathcal{W}}(SZ) \\
&= Z'
\end{aligned}
\tag{22}
$$

$$
\begin{aligned}
P_\pi \mathbf{J} P_\pi^\top &= P_\pi \mathrm{diag}_{i \in [n]} \left( \frac{\partial f_{\mathcal{W}}([SZ]_{i,:})_b}{\partial [SZ]_{i,a}} \right) S P_\pi^\top \\
&= \underbrace{P_\pi \mathrm{diag}_{i \in [n]} \left( \frac{\partial f_{\mathcal{W}}([SZ]_{i,:})_b}{\partial [SZ]_{i,a}} \right) P_\pi^\top}_{(a)} \underbrace{P_\pi S P_\pi^\top}_{=S} \\
&= \mathrm{diag}_{i \in [n]} \left( \frac{\partial f_{\mathcal{W}}([SZ]_{i,:})_b}{\partial [SZ]_{i,a}} \right) S \\
&= \mathbf{J},
\end{aligned}
\tag{23}
$$

where

$$
\begin{aligned}
(a) &= \mathrm{diag}_{i \in [n]} \left( P_\pi \frac{\partial f_{\mathcal{W}}([SZ]_{i,:})_b}{\partial [SZ]_{i,a}} \right) \\
&= \mathrm{diag}_{i \in [n]} \left( \frac{\partial f_{\mathcal{W}}([P_\pi SZ]_{i,:})_b}{\partial [P_\pi SZ]_{i,a}} \right) \\
&= \mathrm{diag}_{i \in [n]} \left( \frac{\partial f_{\mathcal{W}}([SZ]_{i,:})_b}{\partial [SZ]_{i,a}} \right).
\end{aligned}
\tag{24}
$$

$\square$

Then, we prove Proposition 2.

*Proof.* According to the definition of pair-symmetry, for any $(i,j)$, $(k,l)$ within the same equivalence class, i.e. $(i,j) \sim (k,l)$, there exists an automorphism $\pi$ such that $\pi(i) = k$ and $\pi(j) = l$. Then for any $\mathbf{J} \in \mathcal{J}$, according to Lemma 1,

$$
\begin{aligned}
\mathbf{J}_{k,l} &= [P_\pi \mathbf{J} P_\pi^\top]_{k,l} \\
&= [P_\pi \mathbf{J} P_\pi^\top]_{\pi(i),\pi(j)} \\
&= \mathbf{J}_{i,j}.
\end{aligned}
\tag{25}
$$

Meanwhile,

$$
\begin{aligned}
Z'_k &= [P_\pi Z']_k \\
&= [P_\pi Z']_{\pi(i)} \\
&= Z'_i,
\end{aligned}
\tag{26}
$$

and similarly $Z'_l = Z'_j$.

$\square$

## A.6 PROOF OF PROPOSITION 3

*Proof.* We first prove $\text{vec}(\{g_{\boldsymbol{\theta}}(\hat{L})|\boldsymbol{\theta} \in \mathbb{R}^k\}) \subseteq \text{span}\left(\left\{\text{vec}(U_i \otimes U_i)\,\middle|\,i \in [n]\right\}\right)$, where $\hat{L} = U\Lambda U^\top$.

$$
\begin{aligned}
\text{vec}\left(\left\{g_{\boldsymbol{\theta}}(\hat{L})\middle|\boldsymbol{\theta} \in \mathbb{R}^k\right\}\right) &= \text{vec}\left(\left\{Ug_{\boldsymbol{\theta}}(\Lambda)U^\top\middle|\boldsymbol{\theta} \in \mathbb{R}^k\right\}\right) \\
&= \left\{\sum_{i=1}^n g_{\boldsymbol{\theta}}(\boldsymbol{\lambda}_i)\left(\text{vec}(U_i \otimes U_i)\right)\middle|\boldsymbol{\theta} \in \mathbb{R}^k\right\} \\
&\subseteq \left\{\left[\text{vec}(U_i \otimes U_i)\right]_{i\in[n]}\boldsymbol{\alpha}\middle|\boldsymbol{\alpha} \in \mathbb{R}^n\right\} \\
&= \text{span}\left(\left\{\text{vec}(U_i \otimes U_i)\,\middle|\,i \in [n]\right\}\right) \\
&\subset \mathbb{R}^{n^2},
\end{aligned}
\tag{27}
$$

among which $\text{rank}\left(\left\{\text{vec}(U_i \otimes U_i)\,\middle|\,i \in [n]\right\}\right) = \text{rank}(U) = n$.

For a $k$-degree polynomial and $\boldsymbol{\lambda} \in \mathbb{R}^n$,

$$
g_{\boldsymbol{\theta}}(\boldsymbol{\lambda}) = \sum_{i\in[k]} \boldsymbol{\theta}_i \boldsymbol{\lambda}^i = M\boldsymbol{\theta},
\tag{28}
$$

where $\boldsymbol{\theta} \in \mathbb{R}^k$ and $M \in \mathbb{R}^{n\times k}$, $M_{[ij]} = \boldsymbol{\lambda}_i^j$ is a Vandermonde matrix. Hence $M$ is a full rank matrix if all eigenvalues have the algebraic multiplicity equal to 1. If $k = n$, $\{g_{\boldsymbol{\theta}}(\boldsymbol{\lambda})|\boldsymbol{\theta} \in \mathbb{R}^n\} = \{M\boldsymbol{\theta}|\boldsymbol{\theta} \in \mathbb{R}^n\} = \text{span}(M) = \mathbb{R}^n$. Therefore, $\text{vec}\left(\left\{g_{\boldsymbol{\theta}}(\hat{L})|\boldsymbol{\theta} \in \mathbb{R}^k\right\}\right) = \text{span}\left(\{U_i \otimes U_i|i \in [n]\}\right)$.

Next,

$$
\begin{aligned}
\text{vec}(\mathcal{J}_{\boldsymbol{\theta},\mathcal{W}}) &= \text{vec}\left(\left\{\text{diag}_{i\in[n]}\left(\frac{\partial f_{\mathcal{W}}(K_{i,:})_b}{\partial K_{i,a}}\right)g_{\boldsymbol{\theta}}(\hat{L})\middle|\boldsymbol{\theta} \in \mathbb{R}^k, \mathcal{W} \in \mathbb{R}^\star\right\}\right) \\
&= \left\{I \otimes \text{diag}_{i\in[n]}\left(\frac{\partial f_{\mathcal{W}}(K_{i,:})_b}{\partial K_{i,a}}\right)\text{vec}\left(g_{\boldsymbol{\theta}}(\hat{L})\right)\middle|\boldsymbol{\theta} \in \mathbb{R}^k, \mathcal{W} \in \mathbb{R}^\star\right\} \\
&\subseteq \left\{I \otimes \text{diag}(\boldsymbol{\beta})\left[\text{vec}(U_i \otimes U_i)\right]_{i\in[n]}\boldsymbol{\alpha}\middle|\boldsymbol{\alpha}, \boldsymbol{\beta} \in \mathbb{R}^n\right\},
\end{aligned}
\tag{29}
$$

where the last "$\subseteq$" holds as $\left\{\text{diag}_{i\in[n]}\left(\frac{\partial f_{\mathcal{W}}(K_{i,:})_b}{\partial K_{i,a}}\right)\middle|\mathcal{W} \in \mathbb{R}^\star\right\} \subseteq \left\{\text{diag}(\boldsymbol{\beta})\middle|\boldsymbol{\beta} \in \mathbb{R}^n\right\}$ and $\text{vec}\left(\left\{g_{\boldsymbol{\theta}}(\hat{L})|\boldsymbol{\theta} \in \mathbb{R}^k\right\}\right) \subseteq \left\{\left[\text{vec}(U_i \otimes U_i)\right]_{i\in[n]}\boldsymbol{\alpha}\middle|\boldsymbol{\alpha} \in \mathbb{R}^n\right\}$ □

## A.7 PROOF OF PROPOSITION 4

*Proof.* (i) With $K = SZ \in \mathbb{R}^{n\times d}$, for any $\mathcal{J}_{S,\mathcal{W}}|_{S=S_0,\mathcal{W}=\mathcal{W}_0} \in \mathcal{J}_{S,\mathcal{W}}$, we have

$$
\begin{aligned}
\text{vec}\left(\mathcal{J}_{S,\mathcal{W}}|_{S=S_0,\mathcal{W}=\mathcal{W}_0}\right) &= \text{vec}\left(\text{diag}_{i\in[n]}\left(\frac{\partial f_{\mathcal{W}_0}(K_{i,:})_b}{\partial K_{i,a}}\right)S_0\right) \\
&\in \left\{\text{vec}\left(\text{diag}(\boldsymbol{\alpha})S_0\right)\middle|\boldsymbol{\alpha} \in \mathbb{R}^n\right\} \quad /* \text{As } \left[\frac{\partial f_{\mathcal{W}_0}(K_{i,:})_b}{\partial K_{i,a}}\right]_{i\in[n]} \in \mathbb{R}^n */ \\
&\subseteq \bigcup_{S\in\mathcal{S}}\left\{\text{vec}\left(\text{diag}(\boldsymbol{\alpha})S\right)\middle|\boldsymbol{\alpha} \in \mathbb{R}^n\right\}
\end{aligned}
\tag{30}
$$

Therefore, we have

$$
\text{vec}(\mathcal{J}_{S,\mathcal{W}}) \subseteq \bigcup_{S\in\mathcal{S}}\left\{\text{vec}\left(\text{diag}(\boldsymbol{\alpha})S\right)\middle|\boldsymbol{\alpha} \in \mathbb{R}^n\right\}.
\tag{31}
$$

(ii) For any $S \in \mathcal{S}$, we have

$$
\begin{aligned}
\text{vec}(S) &\in \left\{\text{vec}\left(\text{diag}(\boldsymbol{\alpha})S\right)\middle|\boldsymbol{\alpha} \in \mathbb{R}^n\right\} \\
&\subseteq \bigcup_{S\in\mathcal{S}}\left\{\text{vec}\left(\text{diag}(\boldsymbol{\alpha})S\right)\middle|\boldsymbol{\alpha} \in \mathbb{R}^n\right\}.
\end{aligned}
\tag{32}
$$

Hence, $\text{vec}(\mathcal{S}) \subseteq \bigcup_{S\in\mathcal{S}}\left\{\text{vec}\left(\text{diag}(\boldsymbol{\alpha})S\right)\middle|\boldsymbol{\alpha} \in \mathbb{R}^n\right\}$. □

### A.8 PROOF OF PROPOSITION 5

*Proof.* As $\varphi : \mathbb{R}^2 \mapsto \mathbb{R}^d$ is injective, for any $S \in \mathbb{R}^{n \times n \times d'}$, we can find a continuous function $f_S : \mathbb{R}^d \mapsto \mathbb{R}^{d'}$ such that $f_S \circ \varphi((i,j)) = S_{ij}$ for any $i, j \in [n]$. Then according to universal approximation theorem (Cybenko, 1989; Hornik et al., 1989), we can approximate $f_S$ with $f_\Theta$, i.e. there always exists $\Theta = \Theta_0$ such that for any $\epsilon > 0$,

$$\sup_{\boldsymbol{x} \in \mathbb{R}^d} \left\| f_\Theta(\boldsymbol{x})|_{\Theta=\Theta_0} - f_S(\boldsymbol{x}) \right\| < \epsilon. \tag{33}$$

Then

$$
\begin{aligned}
& \sup_{i,j \in [n]} \left\| f_\Theta \circ \varphi((i,j))|_{\Theta=\Theta_0} - S_{i,j,:} \right\| \\
& = \sup_{i,j \in [n]} \left\| f_\Theta \circ \varphi((i,j))|_{\Theta=\Theta_0} - f_S \circ \varphi((i,j)) \right\| \\
& \leq \sup_{\boldsymbol{x} \in \mathbb{R}^d} \left\| f_\Theta(\boldsymbol{x})|_{\Theta=\Theta_0} - f_S(\boldsymbol{x}) \right\| \quad / * \text{ As } \{\varphi(i,j) | i,j \in [n]\} \subseteq \mathbb{R}^d * / \\
& < \epsilon.
\end{aligned}
\tag{34}
$$

$\square$

## B INTERACTION ANALYSIS OF MESSAGE-PASSING NNS

Message-passing framework (MPNN) is a well-adopted concept to generalize various GNN implementations. However, there seems a lack of a unified form since different literatures present it in different forms. They can be summarized into the following two main classes.

$$\boldsymbol{z}_i' = f_\mathcal{V} \left( \boldsymbol{z}_i, \sum_{j=1}^n \hat{A}_{i,j} f_\mathcal{W}(\boldsymbol{z}_j) \right) \tag{35}$$

$$\boldsymbol{z}_i' = f_\mathcal{V} \left( \boldsymbol{z}_i, \sum_{j=1}^n \hat{A}_{i,j} f_\mathcal{W}(\boldsymbol{z}_i, \boldsymbol{z}_j) \right) \tag{36}$$

Equation 35 is studied by Xu et al. (2019); Yang et al. (2022b); Black et al. (2023) etc. Equation 36 is studied by Gilmer et al. (2017); Veličković et al. (2018); Topping et al. (2021) etc. Equation 36 serves as a more general form, and Equation 35 can be viewed as a strict subset of Equation 36. Here, we conduct our interaction efficiency analysis on Equation 36. Equation 36 can be rewritten as the following equivalent form,

$$Z' = f_\mathcal{V} \left( Z, \hat{A} \odot [f_\mathcal{W}(Z_{i,:}, Z_{j,:})]_{i,j \in [n]}^{n \times n \times d} \mathbf{1}^n \right). \tag{37}$$

where $\hat{A} \odot [f_\mathcal{W}(Z_{i,:}, Z_{j,:})]_{i,j \in [n]} \in \mathbb{R}^{n \times n \times d}$, $\odot$ is hadamard product applied on each dimension of $d$ individually. Then we have

$$
\begin{aligned}
Z_{i,b}' &= f_\mathcal{V} \left( Z_{i,:}, \hat{A}_{i,:} \odot [f_\mathcal{W}(Z_{i,:}, Z_{j,:})]_{j \in [n]}^{n \times d} \mathbf{1}^n \right)_b \\
&= f_\mathcal{V} \left( Z_{i,:}, \sum_{j=1}^n \hat{A}_{i,j} f_\mathcal{W}(Z_{i,:}, Z_{j,:})_1, \dots, \sum_{j=1}^n \hat{A}_{i,j} f_\mathcal{W}(Z_{i,:}, Z_{j,:})_d \right)_b.
\end{aligned}
\tag{38}
$$

Then

$$\frac{\partial Z_{i,b}'}{\partial Z_{j,a}} = \frac{\partial f_\mathcal{V}(Z_{i,:}, K_{i,:})_b}{\partial Z_{j,a}} + \sum_{c=1}^d \frac{\partial f_\mathcal{V}(Z_{i,:}, K_{i,:})_b}{\partial K_{i,c}} \hat{A}_{i,j} \frac{\partial f_\mathcal{W}(Z_{i,:}, Z_{j,:})_c}{\partial Z_{j,a}} \in \mathbb{R}, \tag{39}$$

among which $K = \hat{A} \odot [f_\mathcal{W}(Z_{i,:}, Z_{j,:})]_{i,j \in [n]}^{n \times n \times d} \mathbf{1}^n$ and $K_{u,v} = \hat{A}_{u,:} \odot [f_\mathcal{W}(Z_{u,:}, Z_{j,:})_v]_{j \in [n]}^n \mathbf{1}^n$. Then

$$\frac{\partial Z_{i,b}'}{\partial Z_{:,a}} = \frac{\partial f_\mathcal{V}(Z_{i,:}, K_{i,:})_b}{\partial Z_{:,a}} + \sum_{c=1}^d \frac{\partial f_\mathcal{V}(Z_{i,:}, K_{i,:})_b}{\partial K_{i,c}} \left( \hat{A}_{i,:} \odot \left[ \frac{\partial f_\mathcal{W}(Z_{i,:}, Z_{j,:})_c}{\partial Z_{j,a}} \right]_{j \in [n]} \right) \in \mathbb{R}^n. \tag{40}$$

As

$$\frac{\partial f_{\mathcal{V}}(Z_{i,:}, K_{i,:})_b}{\partial Z_{j,a}} = 0 \tag{41}$$

for any $i \neq j$, finally, we have

$$
\begin{aligned}
\mathbf{J}_{\mathcal{W},\mathcal{V}} &= \frac{\partial Z'_{:,b}}{\partial Z_{:,a}} \\
&= \text{diag}_{i\in[n]}\left(\frac{\partial f_{\mathcal{V}}(Z_{i,:}, K_{i,:})_b}{\partial Z_{i,a}}\right) + \\
&\quad \sum_{c=1}^{d}\text{diag}_{i\in[n]}\left(\frac{\partial f_{\mathcal{V}}(Z_{i,:}, K_{i,:})_b}{\partial K_{i,c}}\right)\left(\hat{A}\odot\left[\frac{\partial f_{\mathcal{W}}(Z_{i,:}, Z_{j,:})_c}{\partial Z_{j,a}}\right]_{i,j\in[n]}\right) \\
&\in \mathbb{R}^{n\times n}.
\end{aligned}
\tag{42}
$$

Equation 42 shows some interesting insights into the interaction expressiveness of MPNN:

1. $f_{\mathcal{V}}$ only appears in the computations of the diagonal part of $\mathbf{J}_{\mathcal{W},\mathcal{V}}$. So it has very little contribution to interaction expressiveness, which means designing more sophisticated $f_{\mathcal{V}}$ or increasing the learnable parameters $\mathcal{V}$ will not improve the ability to model complex interactions. $f_{\mathcal{W}}$ appears in the computations of each entry of $\mathbf{J}_{\mathcal{W},\mathcal{V}}$, and therefore the expressiveness of $f_{\mathcal{W}}$ will have a major impact on the interaction expressiveness.
2. MPNN does not well leverage the topologies of the underlying graph in inferring the interactions, as we can see both learnable $f_{\mathcal{W}}$ and $f_{\mathcal{V}}$ only take node features or a pair of node features as inputs with no topology encoding module. And the only topology information is provided by $\hat{A}$. Unfortunately, in most proposed GNNs, $\hat{A}$ is normalized Laplacian or adjacency which only encodes the existence of explicit edges or not. Since most graphs are sparsely connected, most entries in $\hat{A}$ are 0. Then, in each layer computation, the dot-product $\odot$ of $\hat{A}$ masks all pair interactions between nodes with no explicit connections. So, the inappropriate usage of topology acts as an obstacle to interaction computations. The potential improvements towards MPNN can be replacing $\hat{A}$ with learnable $f_{\Theta}\circ\varphi^{\text{LE}}$ as introduced in Section 4.
3. If the applied $\hat{A}_{i,j}$ is invariant to graph automorphism, so does MPNN, i.e. for any automorphism $\pi$ of $G$ with $P_\pi Z = Z$, we have $P_\pi Z' = Z'$ and $P_\pi \mathbf{J} P_\pi^\top = \mathbf{J}$. Then, for any $(i,j)\sim(k,l)$, we have $\mathbf{J}_{i,j} = \mathbf{J}_{k,l}$, $Z'_i = Z'_k$ and $Z'_j = Z'_l$. The proof is similar to that in graph convolution, as in Proposition 2.

## C  SUMMARY OF LOCAL ENCODINGS IN GNNS

Table 4 shows the implementations of $\varphi^{\text{LE}}$ in different GNNs. And their discrimination ability comparisons are shown in Figure 4.

Table 4: Implementations of $\varphi^{\text{LE}}$ in different GNNs, where $\sigma$ is a smoothing function, e.g. $y = e^{\rho\ln x}$, and $P$ is the eigenvector of A.

|  | $\varphi^{\text{LE}}$ |
|---|---|
| GCN, SGC, GPR-GNN, JacobianConv | $[(\tilde{D}^{-\frac{1}{2}}\tilde{A}\tilde{D}^{-\frac{1}{2}})^k]_{k\in[K]}\in\mathbb{R}^{n\times n\times K}$ |
| Spec-GN (Yang et al., 2022a) | $[(P\sigma(\Lambda)P^\top)^k]_{k\in[K]}\in\mathbb{R}^{n\times n\times K}$ |
| PDF (Yang et al., 2023) | $[(\tilde{D}^\epsilon\tilde{A}\tilde{D}^\epsilon)^k]_{k\in[K],\epsilon\in[-0.5,0]}\in\mathbb{R}^{n\times n\times(K|\{\epsilon\}|)}$ |
| GRIT (Ma et al., 2023) | $[(\tilde{D}^{-1}\tilde{A})^k]_{k\in[K]}\in\mathbb{R}^{n\times n\times K}$ |

Apart from existing design of $\varphi^{\text{LE}}$, we provide a new one $[(\tilde{D}^\epsilon\tilde{A}\tilde{D}^{-1-\epsilon})^k]_{k\in[K],\epsilon\in[-1,0]}$. It preserves the benefits of two popular designs while avoiding their respective drawbacks: Compared with $[(\tilde{D}^{-1}\tilde{A})^k]_{k\in[K]}$, it provides more diverse pair encodings, and compared with

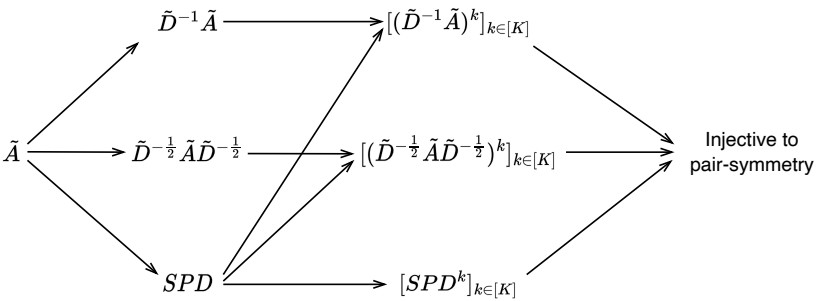

Figure 4: The discrimination ability comparisons of different $\varphi^{\text{LE}}$ implementations, where $[S^k]_{k\in[K]} \in \mathbb{R}^{n\times n\times K}$ refers to the stack of $S^k \in \mathbb{R}^{n\times n}$, and SPD is the shortest path distance matrix with the longest one equal to $K$.

$[(\tilde{D}^\epsilon \tilde{A} \tilde{D}^\epsilon)^k]_{k\in[K],\epsilon\in[-1,0]}$, it has the bounded spectrum thus can be easier to handle numerical instability when applying a larger $K$.

## D EXPERIMENTAL DETAILS.

### D.1 DATASETS STATISTICS.

All detailed statistics of the datasets used in our experiments are presented in Table 5. The corresponding tasks involve graph regression tasks and graph classification tasks collected from real-world molecules, social networks and protein-protein interactions.

Table 5: Statistics of the datasets used in our experiments.

| Dataset | # Graphs | Avg # nodes | Avg # edges | Node attr | Edge attr | Directed | Task type |
|---|---|---|---|---|---|---|---|
| ZINC | 12,000 | 23.2 | 24.9 | Y | Y | N | Regression |
| MNIST | 70,000 | 70.6 | 564.5 | Y | Y | Y | 10-way classi |
| ogbg-molpcba | 437,929 | 26.0 | 28.1 | Y | Y | N | Binary classi. |
| ENZYMES | 600 | 32.63 | 62.14 | Y | N | N | 6-way classi |
| NCI1 | 4110 | 29.87 | 32.39 | N | N | N | Binary classi. |
| NCI109 | 4127 | 29.68 | 32.13 | N | N | N | Binary classi. |
| PTC_MR | 344 | 14.29 | 14.69 | N | N | N | Binary classi. |
| PROTEINS | 1113 | 39.06 | 72.82 | Y | N | N | Binary classi. |
| IMDB-B | 1000 | 19.77 | 96.53 | N | N | N | Binary classi. |
| RDT-B | 2000 | 429.63 | 497.75 | N | N | N | Binary classi. |

### D.2 EXPERIMENTAL SETUP.

Table 6 and Table 7 present all hyperparameter configurations and the number of parameters used in Table 2 and Table 3. We use AdamW (Loshchilov & Hutter, 2018) optimizer. For ZINC and MNIST, we employ a cosine learning rate schedule together with a linear "warm-up" at the beginning of the training.

Table 6: Hyperparameter settings on ZINC, ogbg-molpcba, and MNIST datasets.

| Hyperparameter | ZINC | MNIST | ogbg-molpcba |
|---|---|---|---|
| Hidden Dim. | 160 | 160 | 384 |
| Num. Layers | 6 | 4 | 8 |
| Drop. Rate | 0 | 0 | 0 |
| Readout | mean | mean | max |
| Batch Size | 32 | 16 | 64 |
| Initial LR | 0.001 | 0.001 | 0.0005 |
| LR Dec. Steps | - | - | 5 |
| LR Dec. Rate | - | - | 0.2 |
| # Warm. Steps | 10 | 5 | 5 |
| Weight Dec. | 1e-5 | 1e-5 | 1e-2 |
| # Epochs | 500 | 100 | 15 |
| # Parameters | 499,681 | 110,620 | 3,838,976 |

Table 7: Hyperparameter settings on TUDataset.

| Hyperparameter | ENZYMES | NCI1 | NCI109 | PTC_MR | PROTEINS | IMDB-B | RDT-B |
|---|---|---|---|---|---|---|---|
| Hidden Dim. | 256 | 256 | 256 | 128 | 128 | 256 | 256 |
| Num. Layers | 6 | 6 | 6 | 6 | 6 | 3 | 4 |
| Drop. Rate | 0.2 | 0 | 0 | 0 | 0 | 0 | 0 |
| Readout | max | max | max | max | mean | max | max |
| Batch Size | 64 | 64 | 64 | 64 | 64 | 16 | 64 |
| Initial LR | 0.001 | 0.001 | 0.001 | 0.001 | 0.001 | 0.001 | 0.001 |
| LR Dec. Steps | 40 | 50 | 50 | 30 | 50 | 40 | 50 |
| LR Dec. Rate | 0.6 | 0.6 | 0.6 | 0.65 | 0.65 | 0.6 | 0.6 |
| # Warm. Steps | 0 | 0 | 0 | 0 | 0 | 0 | 0 |
| Weight Dec. | 0 | 0 | 0 | 0 | 0 | 0 | 0 |
| # Epochs | 300 | 300 | 500 | 150 | 300 | 300 | 300 |
| # Parameters | 1,336,070 | 1,335,810 | 1,336,578 | 339,971 | 332,546 | 737,026 | 931,843 |

