# OpenReview forum: "Towards Global Interaction Efficiency of Graph Networks"
_ICLR.cc/2024/Conference — Submitted to ICLR 2024_

### Official Review · Reviewer_uvYG · 2023-10-27

**Soundness:** 2 fair
**Presentation:** 2 fair
**Contribution:** 2 fair
**Rating:** 3
**Confidence:** 4

**Summary:**

In this paper, the authors aim to address the limitations of existing GNNs in representing interactions. They propose to study interactions between node pairs in a graph from a global perspective. They also propose a metric called interaction efficiency for assessing GNN performance. In the analysis of interaction efficiency,  two aspects - interaction sensitivity and interaction expressiveness - are discussed. Finally, a new GNN model, called Universal Interaction Graph Convolution (UIGC), is presented. The authors claim that this proposed GNN model has superior interaction efficiency.

**Strengths:**

Studying interactions among nodes in a graph is of significance to both theoretical foundations and practical applications of the GNN community. Thus, the paper is tackling an important topic. The intention of characterising interactions from the lens of efficiency, sensitivity, and expressiveness also has some novelty. However, the quality of the paper is a concern (see comments in the section "Weaknesses").

**Weaknesses:**

(W1)  The key concepts of the paper (such as global/universal interaction, interaction sensitivity, and interaction expressiveness) are not well-defined.

- The definition of global interaction is defined in terms of the channels of inputs and outputs. How does this relate to interaction patterns, e.g., the five synthetic interaction patterns in Section 5.1? Also, how does this relate to node/pair symmetry?

- For "how a model can universally express any desired interaction within a given graph", what does "universally" mean? The paper also mentions universal interaction but no formal definition is provided.

- Interaction sensitivity is defined to measure the sensitivity of model outputs to perturbations in input node features. However,  perturbations in input node features are not the same as interactions between node pairs - so how are they related? Also, why is  interaction expressiveness considered as one aspect of interaction efficiency?

(W2) The formulation and notations are not well presented.

- Figure 1: What does "a variable capable of taking K values (K=4 there)" mean? How do you decide such a K value?

- What does the notation $S_{i,j,:}$ refer to? What is $\mathbb{R}^*$?

- Equation 6 is not well defined. What do local structures mean precisely? Also the domain of $\varphi^{LE}$ is vague - this needs to be formulated and clearly defined.

- For the statement "the discrimination ability of non-symmetry pairs of $\varphi^{LE}$ is a partial order with the injectivity to be the most discriminative one", it needs a clarification.

- What is the definition of $S/\sim$? Since $S\subseteq \mathbb{R}^{n\times n}$, why is $S/\sim=\mathbb{R}^{\eta}$?

- In Proposition 1, is $f_{\mathcal{W}b}$ a typo? What is $\mathbf{x}_a$?

(W3) Some explanations are needed to improve the clarity of the paper.

- The paper mainly focuses on graph convolution networks as stated in Equation 1. But this is not clarified in the abstract and introduction which seem to consider graph neural networks in general.

- It is unclear how the issues under-reaching and over-squashing mentioned in the abstract and introduction can be addressed by the proposed UIGC layer defined in Equation 7.

- For the statement "UIGC infers the interaction of each pair directly through their local encodings, which will not be affected by the connectivity of graphs", I don't understand this. Why is it not affected by the connectivity of graphs?

(W4) The setup of experiments may cause some confusions. For example,

- How many graphs are randomly selected for the experiments on learning interactions on synthetic data, only one graph? How many graphs are considered in the result presented in Table 1?

- For the experimental results on the five distinct interaction patterns shown in Figure 3, what are these interaction patterns? The current description in Section 5.1 is vague. Also, why are such interaction patterns selected?

- The paper claims that the proposed UIGC can address the issues such as under-reaching and over-squashing of existing works. But there is no experiment provided to compare with existing works on how the proposed UIGC performs for solving such issues.

- How is K selected? Why are only K=3 and K=8 considered?

**Questions:**

See the questions in W1-W4.

---

> ### Author Response · Authors · 2023-11-18
>
> We sincerely thank you for your time and valuable comments.
>
> **(W1) The key concepts of the paper (such as global/universal interaction, interaction sensitivity, and interaction expressiveness) are not well-defined.**
>
> - **The definition of global interaction is defined in terms of the channels of inputs and outputs. How does this relate to interaction patterns, e.g., the five synthetic interaction patterns in Section 5.1? Also, how does this relate to node/pair symmetry?**
>
>   Global interaction as in Eq. 2 defines a $n\times n$ matrix, where each entry corresponds to a node pair. An interaction pattern refers to one such matrix with specific entry values. However, we do not need to deal with all $n\times n$ entries as the symmetric pairs always have the same entry values, which we call **symmetry bias** of GNNs (Please refer to the modification Section 3.2 in the updated PDF). So, let $\eta$ denote the number of symmetric partitions, an interaction pattern is an assignment of $\eta$ partitions. The 5 synthetic interaction patterns in Section 5.1 are 5 different assignments on the $\eta$ partitions.
>
> - **For "how a model can universally express any desired interaction within a given graph", what does "universally" mean? The paper also mentions universal interaction but no formal definition is provided.**
>
>   We refine the statement to "how a model can universally approximate any interaction", as universal approximation is a more well-adopted concept [1]. Then "universal interaction" means a model's capability to approximate any interaction with proper neural parameter assignments. This concept aligns with the idea of universal filters, as explored in various graph signal filtering studies [2, 3, 4, 5]. Please refer to the middle part of Page 2 in the updated PDF for further details.
>
> - **Interaction sensitivity is defined to measure the sensitivity of model outputs to perturbations in input node features. However, perturbations in input node features are not the same as interactions between node pairs - so how are they related? Also, why is interaction expressiveness considered as one aspect of interaction efficiency?**
>
>   As in Eq. 2, the global interaction $\mathrm J_{\theta,W}\in\mathbb R^{n\times n}$ includes interactions among all node pairs ($\mathrm J_{i,j}\in\mathbb R$ indicates the interaction between nodes $i$ and $j$), and it can be formalized as the Jacobian of all node features before and after GNN operations. That is why "perturbations in input node features" and "interactions between node pairs" are related. The determinant of the Jacoabian is commonly used in sensitivity measurement, so we use it to study interaction sensitivity, i.e. the rate of change of outputs with respect to the perturbation of inputs. Briefly, GNNs can learn different interactions $\mathrm J_{\theta,W}$ for the underlying graph, some interactions with small $|\mathrm J_{\theta,W}|$ refer to the change of model output less sensitive to the perturbations of model input.
>   We employ the term 'interaction efficiency' to convey the ability to effectively model diverse and complex interactions. Analogous to filter expressiveness [2, 3, 4, 5], interaction expressiveness gauges the extent to which a model can express/approximate interactions. It serves as the fundamental criterion for interaction efficiency.

---

> > ### Author Response · Authors · 2023-11-18
> >
> > **(W2) The formulation and notations are not well presented.**
> >
> > - **Figure 1: What does "a variable capable of taking K values (K=4 there)" mean? How do you decide such a K value?**
> >
> >   We refine the statement in the caption of Figure 1. Specifically, given a graph, each node can interact with all other nodes, including itself. The interaction states between each node pair can be represented by a variable with a continuous or discrete domain. For a graph with $n$ nodes, assuming there are $K$ discrete interaction states, the number of possible interactions is $K^{\frac{n\times(n+1)}{2}}$. In the provided example, the number of interactions is $4^{28}$.
> >
> > - **What does the notation $S_{i,j,:}$ refer to? What is $\mathbb R^{\star}$?**
> >
> >   For input node features with $d^{\prime}$ channels, $S\in\mathbb R^{n\times n\times d^{\prime}}$ represents all $d^{\prime}$ interactions over these channels. $S_{i,j,:}$ specifically, denotes the interaction across all $d^{\prime}$ channels between the node pair $(i,j)$. When the shape/dimension of the learnable parameter matrix $\mathcal W$ is unknown (as it varies depending on the applied model), we represent it as $\mathcal W\in\mathbb R^{\star}$. We have revised and refined the related statements to mitigate any potential confusion.
> >
> > - **Equation 6 is not well defined. What do local structures mean precisely? Also the domain of $\varphi^{\textrm{LE}}$ is vague - this needs to be formulated and clearly defined.**
> >
> >   $\varphi^{\textrm{LE}}$ represents the computations of encoding graph structures into tensors. As $\varphi^{\textrm{LE}}$ restricts the pair encodings to the mapping of local structures, it produces identical encodings for pairs within the same partitions. The utilization of local structures in GNNs is flexible. For example, when considering $A^k$, the local structures of the pair $(i,j)$ encompass the number of paths with a length of $k$ between nodes $i$ and $j$.
> >
> > - **For the statement "the discrimination ability of non-symmetry pairs of $\varphi^{\textrm{LE}}$ is a partial order with the injectivity to be the most discriminative one", it needs a clarification.**
> >
> >   We refined the related statement in the updated PDF. Specifically, different implementations of $\varphi^{\textrm{LE}}$ exhibit varying discrimination abilities on partitions. For a set of implementations of $\varphi^{\textrm{LE}}$, their discrimination abilities form a partial order. In this partial order, the one injective to the partitions is the most discriminative. Consequently, we consolidate diverse $\varphi^{\textrm{LE}}$ implementations in existing GNNs into a Directed Acyclic Graph (DAG) and compare their discrimination abilities. For detailed information, please refer to Appendix C.
> >
> > - **What is the definition of $S/\sim$? Since $S\subseteq\mathbb R^{n\times n}$, why is $S/\sim\subseteq\mathbb R^{\eta}$?**
> >
> >   We use $S\in\mathbb R^{n\times n}$ to denote a sample interaction among all pairs, and $\mathcal S\subseteq\mathbb R^{n\times n}$ to denote a set of interactions. $\sim$ denotes the equivalence relation induced by pair-symmetry, so $\sim$ can be used to partition all pairs. As pairs within the same partition always have the same interaction as in Proposition 2, we use $\mathcal S/\sim\subseteq\mathbb R^{\eta}$ to denote the set of interactions on partitions, where $\eta$ represents the number of partitions. Then, the universal expressiveness of interactions corresponds to $\mathcal S/\sim=\mathbb R^{\eta}$.
> >
> > - **In Proposition 1, is $f_{\mathcal W b}$ a typo? What is $\mathbf x_a$?**
> >
> >   $f_{\mathcal W b}$ refers to the $b$-th channel of the output of $f_{\mathcal W}$. We fixed some notation issues in the updated PDF.

---

> > > ### Author Response · Authors · 2023-11-18
> > >
> > > **(W3) Some explanations are needed to improve the clarity of the paper.**
> > >
> > > - **The paper mainly focuses on graph convolution networks as stated in Equation 1. But this is not clarified in the abstract and introduction which seem to consider graph neural networks in general.**
> > >
> > >   We also present the results of the MPNN framework in Appendix B due to page limits. The main body primarily focuses on graph convolution since its formulation is more unified, unlike MPNN, which has different formulations in various works. Additionally, our UIGC is an extension of graph convolution.
> > >
> > > - **It is unclear how the issues under-reaching and over-squashing mentioned in the abstract and introduction can be addressed by the proposed UIGC layer defined in Equation 7.**
> > >
> > >   We do not explicitly address the under-reaching or over-squashing issue in long-range interactions. Instead, we demonstrate that this issue is a manifestation of the low expressiveness of interactions. In other words, a model with sufficient expressiveness can effectively handle both long and short-range interactions without the need for explicit specification. The proposed UIGC is designed to achieve such expressive interactions.
> > >
> > > - **For the statement "UIGC infers the interaction of each pair directly through their local encodings, which will not be affected by the connectivity of graphs", I don't understand this. Why is it not affected by the connectivity of graphs?**
> > >
> > >   The interaction inference of UIGC is not influenced by the connectivity of graphs since the inference occurs on the encodings generated by $\varphi^{\textrm{LE}}$. However, more complex connectivities typically demand a more powerful $\varphi^{\textrm{LE}}$ to effectively distinguish pair partitions. For instance, UIGC-8 with 8 bases demonstrates greater power compared to UIGC-3, as discussed in Section 5.1.

---

> > > > ### Author Response · Authors · 2023-11-18
> > > >
> > > > **(W4) The setup of experiments may cause some confusions. For example,**
> > > >
> > > > - **How many graphs are randomly selected for the experiments on learning interactions on synthetic data, only one graph? How many graphs are considered in the result presented in Table 1?**
> > > >
> > > >   We conducted tests on various graphs and obtained consistent results. Both Figure 3 and Table 1 present the results on a randomly selected graph. In practice, our primary focus is to assess the ability of different models to approximate synthetic interactions over partitions.  In this interaction approximation abilities test, graphs are primarily utilized to provide pair partitions.
> > > >
> > > > - **For the experimental results on the five distinct interaction patterns shown in Figure 3, what are these interaction patterns? The current description in Section 5.1 is vague. Also, why are such interaction patterns selected?**
> > > >
> > > >   All five synthetic interactions are functions of the index of pair-symmetry *partitions* rather than pairs, as we need to assign the same interaction value to symmetric pairs within the same partition. These interaction patterns are designed as some typical functions, such as $y_1=const$, $y_2=x$, $y_3=\sin(x)$, etc., each presenting different levels of approximation difficulty. In general, the difficulty of approximation follows the order $y_1\preceq y_2\preceq y_3$. This order is also reflected in Table 1.
> > > >
> > > > - **The paper claims that the proposed UIGC can address the issues such as under-reaching and over-squashing of existing works. But there is no experiment provided to compare with existing works on how the proposed UIGC performs for solving such issues.**
> > > >
> > > >   Our work extends long-range interaction studies, that are considered on individual pairs, to encompass global interactions considered among all pairs simultaneously. It treats long and short-range interactions equally, without the need for specific handling of under-reaching or over-squashing in long-range pairs. Specifically, by applying a $\varphi^{\textrm{LE}}$ being capable of generating distinct encodings for non-symmetric partitions (where long-range and short-range pairs belong to different partitions), the model can infer interactions from the encodings without specifying long-range or short-range pairs. The experimental results demonstrate that a model capable of approximating randomly given global interactions achieves improved predictions.
> > > >
> > > > - **How is K selected? Why are only K=3 and K=8 considered?**
> > > >
> > > >   A larger value of $K$ achieves a more discriminative $\varphi^{\textrm{LE}}$. For the given graph, $K\geq8$ is sufficient to distinguish all pair partitions. In our tests, we evaluate cases with $K=3$ and $K=8$, representing insufficient and sufficient discrimination abilities of $\varphi^{\textrm{LE}}$, respectively. Other values of $K$ can be chosen to demonstrate similar results.
> > > >
> > > >
> > > >
> > > >
> > > >
> > > >
> > > > [1] https://en.wikipedia.org/wiki/Universal_approximation_theorem
> > > >
> > > > [2] He, M., Wei, Z., & Xu, H. (2021). Bernnet: Learning arbitrary graph spectral filters via bernstein approximation. *Advances in Neural Information Processing Systems*, *34*, 14239-14251.
> > > >
> > > > [3] Yang, Mingqi, et al. "A new perspective on the effects of spectrum in graph neural networks." *International Conference on Machine Learning*. PMLR, 2022.
> > > >
> > > > [4] Wang, Xiyuan, and Muhan Zhang. "How powerful are spectral graph neural networks." *International Conference on Machine Learning*. PMLR, 2022.
> > > >
> > > > [5] Bo, Deyu, et al. "Specformer: Spectral Graph Neural Networks Meet Transformers." *The Eleventh International Conference on Learning Representations*. 2022.

---

> > > > > ### Comment · Reviewer_uvYG · 2023-11-23
> > > > >
> > > > > I thank the authors for the detailed responses. I don't have further questions. Nonetheless, some of my major concerns remain. For example, I'm not convinced that the concepts of global/universal interaction and interaction expressiveness are well-defined. In the revised version, the authors mentioned that "Interaction expressiveness shares a similar concern with graph filter expressiveness (Balcilar et al., 2021)". However, the paper by Balcilar et al., 2021 mainly provides some perspective to bridge spectral GNNs and spatial GNNs. There is no specific notion/framework for graph filter expressiveness.

---

> > > > > > ### Author Response · Authors · 2023-11-23
> > > > > >
> > > > > > Please refer to Definition 1 and 2 for the definitions of global interaction and interaction expressiveness respectively.
> > > > > >
> > > > > > Thank you

---

### Official Review · Reviewer_vtTz · 2023-10-31

**Soundness:** 2 fair
**Presentation:** 3 good
**Contribution:** 3 good
**Rating:** 5
**Confidence:** 3

**Summary:**

This paper studies the expressiveness of graph neural networks. In particular, it focuses on the "global efficiency" of GNNs, and the analytical tool this paper proposed is to study the Jacobian matrix of the graph convolution layer.

**Strengths:**

S1. This paper is of good presentation.

S2. The analytic tool proposed by this paper, Jacobian matrix of the graph convolution layer and its determinant, is reasonable.

S3. Experiments of this paper include broad baseline methods and datasets.

S4. This paper is open-sourced to ensure its reproducibility.

**Weaknesses:**

W1. A part of the theory proposed by this paper is problematic.

W2. Some existing contributions are not clearly mentioned, which somehow undermines the contribution of this paper.

W3. Writing in Section 4 can be improved to make the proposed method more clear.

W4. A minor weakness is that the baselines can be selected to only include the most SOTA baselines, and try to ensure most baselines have results on all the datasets (but not only report the results from the original papers)

**Questions:**

Q1. Seems Proposition 5 conflicts with the core idea of this paper. If I understand correctly, the core idea of this paper is "to ensure all symmetric pairs, i.e., within the same pair partitions, learn the same interaction" (from the bottom of Page 5). However, one of the assumptions of Proposition 5 says that $\varphi$ is an injective function, which conflicts with the idea that "symmetric" but different node pairs can learn the same interaction.

Q2.The core design of the proposed UIGC is the $\varphi^{LE}$. However, it is not clearly mentioned what the specific $\varphi^{LE}$ is used in this paper. Table 4 and Figure 4 only show the $\varphi^{LE}$ of existing methods.

Q3. The study on global interaction efficiency is not new. For example, this paper cites [1], which studies “total resistance” $R_{tot}$, a quantitative measure of global interaction efficiency. Thus, the authors should revise the statement of their first contribution. Furthermore, it does not seem necessary to distinguish “global” and “local” interaction efficiency because what the authors mean by “global” is just a function (i.e., determinant, in Section 3.1) of the “local” interaction efficiency matrix ($\mathbf{J}_{\theta,W}$ in Eq. (2)).

Q4. Their UIGC might not be scalable because UIGC needs to compute graph automorphism, which seems to require at least $O(mn^2)$ time. A more efficient approach to addressing the long-distance interaction issue might be increasing the GNN depth together with techniques to alleviate oversquashing (e.g., [2,3] can train GNNs with 7~1000 layers.), or as simple as adding a supernode. Authors should compare with such methods in terms of both efficiency and performance in their experiments.

**Two comments/suggestions:**

C1. The applicable scope of graph automorphism might be limited. A classic result (see, e.g., theorem 2 in [4]) shows that almost all graphs do not have non-trivial automorphism, i.e., in almost all graphs, all but at most one pair of nodes are non-equivalent. In practice, only very special graphs like locally symmetric molecules have non-trivial automorphism. This limitation is also supported by their experiments because their UIGC has at most marginal gain on IMDB and RDT (social network datasets).

C2. The discussion at the end of Section 4 looks vague. It might be more meaningful and more interesting if the authors can extend the discussions to rigorous theoretical analysis.

[1] Black, Mitchell, Zhengchao Wan, Amir Nayyeri, and Yusu Wang. "Understanding oversquashing in gnns through the lens of effective resistance." ICML 2023.

[2] Li et al. DeepGCNs: Can GNNs go as deep as CNNs? ICCV 2019.

[3] Li et al. Training graph neural networks with 1000 layers. ICML 2021.

[4] Erd ̋os & R ́enyi. Assymetric graphs. Acta Math. Acad. Sci. Hungar., 14:295–315, 1963.

---

> ### Author Response · Authors · 2023-11-18
>
> We sincerely thank you for your time and valuable comments.
>
>
> **Q1: Clarifications on Proposition 5**
>
> Proposition 5 serves as a general and self-consistent conclusion, with no need to specify symmetric or non-symmetric cases. Specifically, for the symmetric case, although injective $\varphi$ will generate different outputs for symmetric pairs, the MLP $f_{\Theta}$ which takes the outputs of $\varphi$ as inputs can learn the same interaction for them. So Proposition 5 always holds, whether considering the symmetry or not. Then, to ensure symmetric pairs always learn the same interactions, we relax the injectivity of $\varphi$ by ensuring that symmetric pairs always share the same output as in Equation 6.
>
> We note that the related statement may result in such confusion, and we update the related statements in the updated PDF to make it clearer.
>
>
> **Q2: What is the specific $\varphi^{\textrm{LE}}$ is used in this paper?**
>
> Apart from existing design of $\varphi^{\textrm{LE}}$, we propose the new one $[(\tilde D^{\epsilon}\tilde A\tilde D^{-1-\epsilon})^k]\_{k\in[K],\epsilon\in[-1,0]}$. It preserves the benefits of two popular designs while avoiding their respective drawbacks: Compared with $[(\tilde D^{-1}\tilde A)^k]\_{k\in[K]}$ in [1], it provides more diverse pair encodings, and compared with $[(\tilde D^{\epsilon}\tilde A\tilde D^{\epsilon})^k]\_{k\in[K],\epsilon\in[-1,0]}$ in [2], it has the bounded spectrum thus can be easier to handle numerical instability when applying a larger $K$.
>
> Given the flexible nature of the $\varphi^{\textrm{LE}}$ design, our primary focus lies in understanding its general properties, such as the partial order among implementations and the presence of symmetry bias. As a result, we include our implementation of $\varphi^{\textrm{LE}}$ in the appendix rather than the main body of the updated PDF.
>
>
>
>
> **Q3: Comparisons with “total resistance” [3] & Possible misunderstandings on global interactions**
>
> Our interaction efficiency analysis involves interaction sensitivity and interaction expressiveness respectively. The former one may be comparable to "total resistance" $R_{tot}$ [3] as they all measure the property of the entire graph with a scalar. But the considered metrics are different. We use the determinant of the Jacobian of all nodes. It is a commonly used metric in stability and sensitivity analysis, and in our settings, it captures the rate of change of all output nodes to changes in all input nodes. The total resistance study extends local pairwise effective resistance to the global scenario by summing of the effective resistance between all pairs of nodes.
>
> "Furthermore, it does not seem necessary to distinguish 'global' and 'local' interaction efficiency because ..." seems to be a misunderstanding. Here is the clarification:
>
> Our work aims to extend existing individual pair interaction studies, e.g. long-range interaction, to the global scenario. To this end, we use $\mathrm J_{\theta,W}\in\mathbb R^{n\times n}$ in Eq. 2 to model global interactions. Based on $\mathrm J_{\theta,W}$, we analyze
>
> - interaction sensitivity, measured by its determinant $|\mathrm J_{\theta,W}|$ in Section 3.1, and
> - interaction expressiveness, measured by the space $\{\mathrm J_{\theta,W}\big|\theta\in\mathbb R^k,W\in\mathbb R^{\star}\}$ in Section 3.2.
>
> We added discussions on the difference with [3] in Section 1 in the updated PDF.
>
>
>
> **Q4: The scalability of UIGC**
>
> Our UIGC does not require the computation of graph automorphisms. Notably, learning the same interactions for symmetric pairs (where symmetry is induced by graph automorphisms) is inherently satisfied by all GNNs, including ours. This is facilitated by the **symmetry bias** inherent in $\varphi^{\textrm{LE}}$, as discussed in Section 4. The mention of graph automorphisms in our submission is solely for the purpose of interaction expressiveness analysis and does not introduce any additional computations.

---

> > ### Author Response · Authors · 2023-11-18
> >
> > **C1: The applicable scope of graph automorphism might be limited.**
> >
> > Thanks for the paper you mentioned which presents results on random graphs. Regular graphs, typically sparsely connected, are more prone to having non-trivial automorphisms. For instance, a graph with a node having two non-adjacent neighbors, and these two neighbors having no other neighbors, allows us to swap these two neighbors to create a non-trivial automorphism. In contrast, asymmetric graphs are rare for regular graphs. For instance, the smallest asymmetric graphs have 6 nodes, and the smallest asymmetric regular graphs have 10 nodes, etc. [4]
> >
> >
> > **C2: The discussion at the end of Section 4 looks vague.**
> >
> > Some discussions are provided in Appendix B and C.
> >
> >
> >
> > [1] Ma, Liheng, et al. "Graph Inductive Biases in Transformers without Message Passing."
> >
> > [2] Yang, Mingqi, et al. "Towards Better Graph Representation Learning with Parameterized Decomposition & Filtering."
> >
> > [3] Black, Mitchell, Zhengchao Wan, Amir Nayyeri, and Yusu Wang. "Understanding oversquashing in gnns through the lens of effective resistance."
> >
> > [4] https://en.wikipedia.org/wiki/Asymmetric_graph

---

> > > ### Comment · Reviewer_vtTz · 2023-12-03
> > > **Acknowledgement to authors' response**
> > >
> > > I thank the authors' response and revision. Personally, I did not fully understand the rationale of studying Proposition 5 (e.g., "it is not related to symmetric or non-symmetric cases" then why mention it here) and still thought that it is not closely related to the proposed solution. In addition, I admit that the proposed metric of this paper is different from the “total resistance” from [3], but I still believe both metrics can be used to measure "global interaction efficiency". For the other parts, thanks for the clarification. Overall, my evaluation for this paper is borderline, and might be slightly below the borderline. Thus, I am keeping my current score.

---

### Official Review · Reviewer_6p3F · 2023-11-03

**Soundness:** 3 good
**Presentation:** 2 fair
**Contribution:** 3 good
**Rating:** 6
**Confidence:** 2

**Summary:**

In this paper, the authors aim to investigate the global interactions in graphs. In particular, they propose a metric named global efficiency for accessing GNNs' performance. It moves beyond the traditional approach of focusing on local interactions between individual node pairs and examines GNN interactions from a global perspective. Furthermore, inspired by the insights from these investigations, they propose Universal Interaction Graph Convolution (UIGC) with very superior interaction efficiency.

**Strengths:**

*Originality*: The paper is original in its approach to a fundamental challenge in the field of GNNs. It introduces the novel concept of universal interaction expression within graphs, surpassing conventional limitations. The proposed solution to utilize the Jacobian matrix for quantifying interaction efficiency, distinguishing between interaction sensitivity and expressiveness, is original and useful.


*Significance*: The paper has the potential to enhance the capabilities of GNNs from a new perspective.

**Weaknesses:**

*Motivation*: It would be helpful if the authors could provide better motivations for some of their arguments and designs. For instance, throughout the paper, the authors are discussing about pair interactions. However, it is not very clearly demonstrated why they are so important.

*Experiments*: The results in Table 2 and Table 3 show that the proposed method achieves strong performance on some of the datasets while falling short on some others. It would be helpful if the authors could provide more explanations on why this is the case. Some investigations into the data properties and their connections to global interactions would be especially helpful.

**Questions:**

Please address the questions in the Section of weakness. In addition, there are a few other questions as follows.

1. In Figure 1, why do we specifically care about pairs of nodes and their interactions? It would be better if the authors could provide more details and better descriptions.

2. It might be helpful if the authors could provide some examples on how the proposed method is more expressive in terms of capturing patterns such as "benzene rings" (mentioned in the introduction).

3. The setup for Section 5.1 is not very clear. It would be helpful if the authors could provide more descriptions of the synthetic interaction patterns. In particular, what is the value in the y-axis in Figure 3?

---

> ### Author Response · Authors · 2023-11-18
>
> We sincerely thank you for your time and valuable comments.
>
> **W1, Q1, Q2: Why do we specifically care about pairs of nodes and their interactions? & Why do we study global interactions?**
>
> - Why do we specifically care about pairs of nodes and their interactions?
>
>   In a graph, one node can influence another node through direct or indirect connections. Pair interactions serve as a fundamental component to model this mutual influence. For instance, existing studies on long-range interactions concentrate on scenarios where a pair of nodes is distant from each other, demonstrating that their pair interaction decays exponentially with respect to the distance [1, 2].
>
> - Why we should study global interactions which consider all pair interactions simultaneously?
>
>   The influence among distinct pairs of nodes is not independent. Certain information can only be effectively captured when examined in a 'simultaneous' rather than an 'individual' manner. To explain this, we replace the "benzene rings" example with a more intuitive one: considering the scenario where A, B, and C co-author a paper together, where the relationship cannot be accurately represented by breaking it down into three separate relations like 'A and B co-author a paper together,' 'B and C co-author a paper together,' and 'A and C co-author a paper together.' The reason is that these individual relations also encompass cases where A&B, B&C, and A&C co-author in three distinct papers, respectively. To address this, we adopt a global interactions perspective, where we consider all node pairs and their interactions simultaneously, as illustrated in Figure 1.
>
> Interaction expressiveness shares a similar concern with graph filter expressiveness [3]. For a graph with $n$ nodes, The corresponding Laplacian involves $n$ graph Fourier bases, each associated with a filtering coefficient. The concurrent consideration of filtering across different bases results in a filter space denoted as $\mathbb R^n$ [4, 5, 6, 7]. Recent studies emphasize the necessity to enhance filter expressiveness to effectively span and cover this space. In a parallel analogy, within the framework of global interactions, each node pair is assigned an interaction coefficient. To view all pair interactions simultaneously, the dimensions of the interaction space is related to the number of pairs.
>
> We have refined the description of our motivations in Section 1 in the updated PDF.
>
>
> **W2: Experiments: The results in Table 2 and Table 3 show that the proposed method achieves strong performance on some of the datasets while falling short on some others.**
>
> In UIGC, we utilize the MLP $f_{\Theta}$ to model interactions. The classification improvement on small-scale datasets shows the alleviation of the overfitting issue. However, datasets like IMDB-B have no classification gains, indicating that label-related interaction patterns on these graphs may be inherently simple and can be easily captured by basic models.
>
>
> **Q3: The setup for Section 5.1**
>
> We refined the description of the setup in Section 5.1 in the updated PDF. In Section 5.1, we test five interaction patterns with different approximation difficulties, including $h(k)=\alpha$,  $h(k)=\alpha k$, $h(k)=k^{\alpha}$, $h(k)=\sin(\alpha k)$ and $h(k)=\alpha\lceil k/40\rceil$, where $k$ is the pair partition index, $h(k)$ is the synthetic interaction, $\alpha$ is a scalar used to control the output. In Figure 3, x-axis is the partition index $k$ and y-axis is the interaction $h(k)$.
>
> [1] Topping, Jake, et al. "Understanding over-squashing and bottlenecks on graphs via curvature." International Conference on Learning Representations. 2021.
>
> [2] Liu, Juncheng, et al. "Mgnni: Multiscale graph neural networks with implicit layers." Advances in Neural Information Processing Systems 35 (2022): 21358-21370.
>
> [3] Balcilar, Muhammet, et al. "Analyzing the expressive power of graph neural networks in a spectral perspective." Proceedings of the International Conference on Learning Representations (ICLR). 2021.
>
> [4] He, M., Wei, Z., & Xu, H. (2021). Bernnet: Learning arbitrary graph spectral filters via bernstein approximation. *Advances in Neural Information Processing Systems*, *34*, 14239-14251.
>
> [5] Yang, Mingqi, et al. "A new perspective on the effects of spectrum in graph neural networks." *International Conference on Machine Learning*. PMLR, 2022.
>
> [6] Wang, Xiyuan, and Muhan Zhang. "How powerful are spectral graph neural networks." *International Conference on Machine Learning*. PMLR, 2022.
>
> [7] Bo, Deyu, et al. "Specformer: Spectral Graph Neural Networks Meet Transformers." *The Eleventh International Conference on Learning Representations*. 2022.

---

### Author Response · Authors · 2023-11-22
**Request for feedback from Reviewers**

Dear Reviewers,

We thank all reviewers for your constructive comments. Please check our rebuttal, and please let us know if you have any further questions.

Sincerely,

Authors

---

### Meta-Review · Area_Chair_itmH · 2023-12-07

**Metareview:**

The paper proposes defining and modeling  global interactions for graph neural networks using the Jacobian of the node features, and characterize two interaction dimensions - interaction expressiveness and interaction sensitivity. The authors use this to motivate a new GNN and evaluate it on multiple datasets and baselines.

The authors are addressing an important topic in graph neural networks and the idea of utilizing the Jacobian is reasonable. However reviewers expressed concerns about the clarify and presentation of the ideas in the paper, which undermined its contributions. In particular, their central notions of separately characterizing interaction sensitivity and interaction expressiveness seems forced, and somewhat  poorly motivated and justified given that they ultimately are artifacts of the same Jacobian analysis. Another concern raised was around insufficiently comparing this work against previous papers that also studied different techniques for incorporating global interactions including effective resistance and oversquashing.

Nevertheless, I think the empirical results in this paper are interesting - I hope the authors consider revising the paper with a better treatment and presentation of their global interaction notions.

**Justification For Why Not Higher Score:**

Despite the somewhat reasonable empirical results in the paper, there is unnecessary complexity, poor presentation and overclaiming in this paper around global interaction concepts that go beyond what is essentially a simple idea of using the Jacobian, which the reviewers pointed out and which I tend to agree with.

**Justification For Why Not Lower Score:**

N/A

---

### Decision · Program_Chairs · 2024-01-16

Reject